# Shaping the Future of Healthcare: Ethical Clinical Challenges and Pathways to Trustworthy AI

**DOI:** 10.3390/jcm14051605

**Published:** 2025-02-27

**Authors:** Polat Goktas, Andrzej Grzybowski

**Affiliations:** 1UCD School of Computer Science, University College Dublin, D04 V1W8 Dublin, Ireland; polat.goktas@ucd.ie; 2Department of Ophthalmology, University of Warmia and Mazury, 10-719 Olsztyn, Poland; 3Institute for Research in Ophthalmology, Foundation for Ophthalmology Development, 61-553 Poznan, Poland

**Keywords:** artificial intelligence, bias, ethics, health policy, machine learning, natural language processing, large language model, privacy, regulation

## Abstract

**Background/Objectives**: Artificial intelligence (AI) is transforming healthcare, enabling advances in diagnostics, treatment optimization, and patient care. Yet, its integration raises ethical, regulatory, and societal challenges. Key concerns include data privacy risks, algorithmic bias, and regulatory gaps that struggle to keep pace with AI advancements. This study aims to synthesize a multidisciplinary framework for trustworthy AI in healthcare, focusing on transparency, accountability, fairness, sustainability, and global collaboration. It moves beyond high-level ethical discussions to provide actionable strategies for implementing trustworthy AI in clinical contexts. **Methods**: A structured literature review was conducted using PubMed, Scopus, and Web of Science. Studies were selected based on relevance to AI ethics, governance, and policy in healthcare, prioritizing peer-reviewed articles, policy analyses, case studies, and ethical guidelines from authoritative sources published within the last decade. The conceptual approach integrates perspectives from clinicians, ethicists, policymakers, and technologists, offering a holistic “*ecosystem*” view of AI. No clinical trials or patient-level interventions were conducted. **Results**: The analysis identifies key gaps in current AI governance and introduces the *Regulatory Genome*—an adaptive AI oversight framework aligned with global policy trends and Sustainable Development Goals. It introduces quantifiable trustworthiness metrics, a comparative analysis of AI categories for clinical applications, and bias mitigation strategies. Additionally, it presents interdisciplinary policy recommendations for aligning AI deployment with ethical, regulatory, and environmental sustainability goals. This study emphasizes measurable standards, multi-stakeholder engagement strategies, and global partnerships to ensure that future AI innovations meet ethical and practical healthcare needs. **Conclusions**: Trustworthy AI in healthcare requires more than technical advancements—it demands robust ethical safeguards, proactive regulation, and continuous collaboration. By adopting the recommended roadmap, stakeholders can foster responsible innovation, improve patient outcomes, and maintain public trust in AI-driven healthcare.

## 1. Introduction

Artificial intelligence (AI) is reshaping the global healthcare landscape, offering transformative opportunities for diagnostics, treatment, and patient care. The rapid growth of AI technologies, fuelled by advancements in machine learning (ML), digitized health data, and computational power, has enabled breakthroughs in areas traditionally dominated by human expertise [1,2,3,4]. This article presents a structured synthesis of the relevant literature, integrating insights from multiple disciplines, including healthcare, ethics, policy, and AI governance. Recognizing that achieving reliable AI is inherently interdisciplinary, we explicitly incorporate perspectives from law, sociology, computer science, and public health to offer a holistic analysis of AI’s implications in healthcare. Our discussion outlines how interdisciplinary collaboration among ethicists, clinicians, AI engineers, legal experts, and policymakers can shape AI governance and mitigate risks. To ensure a comprehensive perspective, sources were selected using a targeted review process, prioritizing peer-reviewed journal articles in PubMed, Scopus, and Web of Science published within the last decade. The selection criteria focused on studies that critically evaluate AI’s ethical, regulatory, and implementation challenges, allowing us to present a holistic view of the field.

From predicting disease trajectories to optimizing resource allocation, AI has the potential to address long-standing challenges in healthcare, enhancing precision, efficiency, and accessibility [5,6,7,8]. However, the existing literature often examines AI’s technological advancements in isolation, overlooking the intersection of ethical risks, regulatory gaps, and real-world implementation challenges. This fragmented approach creates a critical research gap, as there is a lack of comprehensive, multidisciplinary frameworks that integrate AI ethics, governance, and practical deployment strategies in healthcare. Furthermore, different categories of AI systems vary in their interpretability, risk profiles, and suitability for healthcare applications [9]. While traditional rule-based AI models and decision trees offer greater transparency, deep learning-based AI—especially neural networks—often lacks interpretability but outperforms simpler models in tasks such as medical image analysis and genomics. Explainable AI (XAI) techniques are emerging as a bridge, seeking to balance accuracy and interpretability in clinical applications [10,11]. However, there is still a pressing need to assess whether certain AI categories provide inherent advantages over others in terms of safety, accountability, and regulatory feasibility.

In addition to ethical and legal concerns, AI-driven healthcare transformations have profound social consequences. The accessibility of AI-powered medical interventions raises concerns about the digital divide—patients in low-income regions or marginalized communities may face disproportionate barriers to benefiting from AI-assisted healthcare. Limited digital literacy, infrastructure deficits, and algorithmic biases can reinforce existing health disparities rather than alleviate them [12]. Furthermore, the automation of certain healthcare processes raises questions about the displacement of healthcare workers, requiring strategic workforce adaptation to ensure AI complements rather than replaces human expertise. The societal impact of AI in healthcare extends to patient autonomy, where algorithmic decision-making could influence treatment pathways in ways that may challenge shared decision-making models between patients and physicians. Without careful oversight, AI-driven clinical tools could exacerbate power imbalances, particularly among populations with limited healthcare agencies or lower health literacy [13]. This study addresses this gap by providing a cross-disciplinary perspective on AI governance, offering concrete recommendations for ensuring transparency, fairness, and sustainability in AI-driven healthcare systems. As AI continues to evolve, bridging these gaps is essential to maximizing its benefits while mitigating ethical, legal, and societal risks.

### 1.1. The Promise and Risks of AI in Healthcare

AI’s expansion into healthcare presents unique risks that necessitate robust ethical frameworks. Patient safety, data privacy, and regulatory uncertainty are major concerns. In clinical applications, AI models often function as opaque “*black boxes*”, making it difficult for healthcare professionals to interpret or trust their decision-making processes [14,15,16]. For example, convolutional neural networks (CNNs) used in medical imaging, such as AI-powered radiology tools for detecting tumors in MRI scans, often lack clear explanations for their predictions, making it difficult for healthcare professionals to interpret or trust their decision-making processes [17,18]. Similarly, AI-driven sepsis prediction models in intensive care units have demonstrated high accuracy but provide little insight into the reasoning behind their risk assessments, leading to concerns over clinical adoption [19].

In selecting the references for this section, we prioritized studies that examine real-world challenges in AI governance, data privacy, and the limitations of current ethical frameworks. We also included policy analyses that assess gaps in existing regulations and explore potential governance solutions. This opacity not only hinders trust but also raises concerns about accountability when errors occur. A key issue is the need for explainable and interpretable AI models. Without transparency, AI decisions may undermine clinician trust and accountability [20,21,22]. Additionally, AI applications rely heavily on vast amounts of patient data, creating privacy concerns and exposing vulnerabilities to cyberattacks [23]. The ethical use of such sensitive information requires adherence to principles of non-maleficence and beneficence, aligning with the core tenets of medical ethics. Regulatory frameworks often lag behind technological advancements, leading to a fragmented landscape that varies across regions, further complicating the adoption of AI in healthcare [24,25]. These regulatory gaps emphasize the importance of establishing harmonized global standards to ensure that AI technologies are deployed responsibly and equitably.

Beyond these ethical and regulatory challenges, AI in healthcare also introduces significant social implications. The integration of AI into healthcare workflows may lead to shifts in physician-patient interactions, altering traditional models of care delivery. As AI systems become more prevalent in diagnostics and treatment planning, the human element of medical decision-making could be diminished, affecting patient experiences and trust in healthcare providers. Additionally, the employment landscape in healthcare is evolving due to automation, necessitating new training initiatives to prepare healthcare workers for AI-enhanced roles rather than complete job displacement [26]. These shifts require careful policymaking to ensure AI adoption enhances rather than disrupts healthcare accessibility, workforce stability, and equitable patient outcomes.

### 1.2. Instances of AI Misuse in Healthcare and Their Repercussions

The misuse of AI, especially for generative AI tools such as ChatGPT 4o or o1 versions, in healthcare, poses significant risks, including falsification of medical records, misinformation, algorithmic bias, and privacy violations.

**Fabrication of Medical Records:** While synthetic data generation aids AI training, the introduction of falsified medical records into clinical workflows or fraudulent claims can mislead clinicians and distort research outcomes [27].**Misinformation in Medical Advice:** AI-generated medical guidance can be erroneous, as seen in studies where chatbots provided incorrect cancer treatment recommendations with fabricated sources, endangering patient safety [28].**Algorithmic Bias Amplification:** AI models trained on non-representative datasets have perpetuated disparities, such as dermatological AI misdiagnosing conditions in patients with darker skin tones because of biased training data [29].**Deepfake Medical Content:** AI-generated videos and images have been exploited to spread false health information, such as vaccine misinformation, fueling public distrust and hesitancy [30].

The improper use of generative AI undermines patient trust, compromises clinical decision-making, and creates legal liabilities. Cybersecurity risks, including identity fraud and AI-powered phishing attacks, further threaten healthcare systems. Addressing these challenges requires stringent regulatory oversight, routine AI auditing, and interdisciplinary collaboration among AI developers, healthcare professionals, and policymakers to ensure ethical and responsible deployment [31].

### 1.3. Defining Trustworthy AI

Ensuring the ethical deployment of AI in healthcare is not just about mitigating risks but also about establishing a foundation of trust. This leads us to the concept of Trustworthy AI, a fundamental pillar for its successful integration into healthcare, as illustrated in Figure 1. This concept encompasses key principles such as transparency, accountability, fairness, and patient autonomy [16,32]. Our review included empirical studies evaluating how AI models incorporate these principles and regulatory reports outlining best practices:**Transparency** ensures AI decisions are comprehensible.**Accountability** requires developers and users to be responsible for outcomes.**Fairness** addresses bias in AI algorithms, ensuring equitable healthcare access.**Patient autonomy** ensures AI empowers patients rather than limiting decision-making.

Beyond these core principles, other essential dimensions, such as safety, security, and robustness, must also be considered. Safety ensures AI-driven systems prioritize patient well-being and do not introduce new risks in clinical settings [33]. Security is crucial in preventing cyber threats and protecting sensitive health data from unauthorized access or manipulation [34]. Robustness refers to an AI system’s ability to perform reliably across diverse populations and medical conditions, avoiding failures due to adversarial attacks or unforeseen data variations [35].

Achieving trustworthy AI is not just a technical issue—it demands collaboration among engineers, ethicists, healthcare providers, and policymakers to address these additional dimensions and ensure AI systems function reliably under all conditions [36,37]. This interdisciplinary approach is necessary to balance technological feasibility with ethical integrity, regulatory compliance, and clinical applicability. For instance, integrating explainability mechanisms in AI models requires input from both computer scientists and healthcare professionals to ensure usability in medical decision-making. Importantly, AI must be designed to operate within socially responsible frameworks, ensuring it benefits diverse populations rather than exacerbating existing health disparities. The risk of algorithmic bias disproportionately affecting minority or low-income communities remains a pressing issue, requiring proactive bias mitigation strategies at the design stage to prevent inequitable healthcare outcomes while maintaining security, safety, and resilience in real-world settings [5].

### 1.4. AI as an Ecosystem

AI does not function in isolation—it operates within a complex healthcare ecosystem, integrating into clinical workflows, data infrastructures, and regulatory environments [23]. An AI ecosystem perspective highlights the importance of interoperability, data standardization, and collaboration among stakeholders to maximize the potential of AI technologies. Our literature selection prioritized studies on AI interoperability, data governance, and human–AI interaction. Moreover, the dynamic interplay between AI systems and human users—whether clinicians, patients, or administrators—demands a focus on human–AI interaction to ensure that these systems enhance rather than undermine healthcare delivery [2,14].

The regulatory landscape is a critical component of this ecosystem. Emerging policies, such as the European Union (EU)’s Artificial Intelligence Act (AIA), aim to establish stringent requirements for high-risk AI systems, including those used in healthcare [38,39]. These regulations emphasize the need for transparency, risk management, and compliance mechanisms to safeguard patient safety and public trust [32]. However, the global diversity in regulatory approaches presents challenges for developers and healthcare institutions operating across borders, necessitating international cooperation and harmonization—a key theme in our proposed Regulatory Genome framework.

Additionally, the social and ethical implications of AI as an integrated ecosystem must be accounted for. The rise of AI-driven digital health platforms and telemedicine solutions has reshaped traditional patient care models, raising concerns about digital exclusion for elderly patients, rural populations, and those with limited technological literacy. Ensuring AI-driven healthcare remains equitable and accessible to all patients—regardless of socioeconomic status—requires targeted interventions, regulatory oversight, and ethical design principles [40,41]. This ecosystem approach informs our later discussion on bias mitigation, fairness, and sustainability, reinforcing the need for structured accountability mechanisms that extend beyond individual AI models to the broader systems in which they operate.

### 1.5. Towards a Regulatory Genome

The concept of a “*regulatory genome*” reflects the need for a comprehensive and adaptive framework that governs the development, deployment, and monitoring of AI systems in healthcare. This framework should integrate ethical principles, technical standards, and legal requirements, ensuring that AI technologies align with societal goals and healthcare priorities [24]. Key components include:**Continuous monitoring and evaluation** of AI performance.**Stakeholder engagement** to ensure diverse perspectives are included.**Frameworks to address emerging risks** such as adversarial attacks and AI misuse [21].

The regulatory genome also aligns AI development with the United Nations Sustainable Development Goals (SDGs) by proposing concrete assessment and implementation methods to ensure AI-driven healthcare solutions promote equity and accessibility [24,37]. For instance, AI’s role in predictive disease modeling can enhance early intervention strategies, reducing healthcare costs while improving patient outcomes. However, if not properly governed, AI innovations may contribute to systemic inequities, reinforcing existing disparities rather than reducing them.

To move beyond a theoretical construct, we introduce a hypothetical case study demonstrating the practical implementation of the regulatory genome in healthcare:

**Case Study:** AI-Driven Early Detection of Diabetic Retinopathy in Low-Resource Settings A global health initiative deploys an AI-powered diabetic retinopathy screening system in rural areas where access to ophthalmologists is limited. The regulatory genome framework can enable that:**Transparency**: The AI model’s decision-making process is explainable through heatmaps and confidence scores, helping clinicians validate its outputs.**Bias Mitigation**: The model can undergo algorithmic fairness audits, ensuring it does not disproportionately misdiagnose patients from underrepresented ethnic groups.**Compliance**: The AI system should adhere to the EU AI Act’s high-risk category standards and follow data protection regulations such as GDPR, with federated learning techniques ensuring privacy-preserving AI training.**Continuous Monitoring**: The system can undergo quarterly post-market surveillance audits, assessing real-world performance across diverse populations.**Stakeholder Involvement**: Regulators, clinicians, and local community representatives contribute to policy adjustments, ensuring culturally sensitive AI deployment.

By operationalizing this adaptive model, the regulatory genome helps keep AI in healthcare transparent, fair, and responsive to changing societal needs.

### 1.6. Translating the AI Ecosystem Perspective into Actionable Policy and Governance Strategies

To effectively assess and implement SDG alignment, we propose the following key methods:**Impact Assessment Metrics**: Developing standardized AI impact indicators aligned with specific SDGs, such as healthcare accessibility (SDG 3), reduced inequalities (SDG 10), and responsible innovation (SDG 9). A study by Vinuesa et al. (2020) provides a comprehensive review of how AI can act as an enabler across various SDG targets, highlighting the importance of creating metrics to assess AI’s contributions to these goals [24].**AI Governance Audits**: Requiring AI systems in healthcare to undergo periodic audits to evaluate their contributions to sustainability, ethical compliance, and fairness [42].**Regulatory Incentives**: Encouraging AI developers and healthcare institutions to adopt SDG-aligned AI practices through regulatory incentives, such as funding opportunities and compliance certifications [43].**Stakeholder-driven Evaluation**: Engaging multidisciplinary experts, including healthcare professionals, ethicists, and policymakers, in qualitative and quantitative reviews of AI’s contributions to sustainable healthcare ecosystems. A systematic literature review by Birkstedt et al. (2023) emphasizes the importance of involving diverse stakeholders in AI governance to ensure comprehensive evaluation and accountability [44].

While the promise of AI in healthcare is undeniable, the preceding analysis highlights critical gaps in addressing its ethical, regulatory, and practical challenges. The current literature often emphasizes either the technological advancements or the ethical principles in isolation, leaving a significant void in actionable frameworks that integrate these aspects comprehensively. This paper addresses that gap by offering a unified approach to understanding and implementing trustworthy AI in healthcare. Our focus stems from the urgent need to navigate the complexities of privacy, bias, regulatory ambiguity, and patient autonomy, which remain underexplored in their interplay with real-world clinical perspectives. By bridging theoretical principles with practical implementation, this paper provides measurable frameworks and policy recommendations tailored to the evolving healthcare ecosystem. Moreover, it emphasizes emerging challenges such as adversarial resilience and sustainable AI development, areas that are critical but insufficiently discussed in the existing literature. For example, adversarial attacks can manipulate medical AI systems by subtly altering input data, leading to incorrect diagnoses or treatment recommendations [45]. Addressing this requires developing robust algorithms capable of detecting and mitigating such attacks. Additionally, sustainable AI development involves creating models that are not only effective but also environmentally responsible, considering factors such as energy consumption and resource utilization [46]. This paper not only advances academic discourse but also equips stakeholders with the tools to ensure ethical AI integration, fostering improved patient outcomes and equitable healthcare delivery.

### 1.7. Key Contributions of This Study

This paper provides a structured, interdisciplinary framework that bridges AI ethics, governance, and real-world deployment strategies in healthcare. The main contributions are as follows:**A Unified AI Governance Framework:** Unlike existing approaches that focus separately on AI ethics or regulation, this study integrates transparency, fairness, accountability, and sustainability into a cohesive governance structure.**Quantifiable AI Trustworthiness Metrics:** We introduce measurable indicators—such as explainability scores, bias reduction rates, compliance metrics, and sustainability benchmarks—to evaluate AI systems in healthcare.**Comparative Analysis of AI Categories:** We assess symbolic AI, ML, and hybrid AI approaches, identifying their strengths, risks, and suitability for clinical applications.**Regulatory Genome Concept for AI Oversight:** We introduce an adaptive regulatory framework that aligns with global AI governance trends and SDGs, ensuring ethical AI deployment.**Bias Mitigation and Algorithmic Fairness Strategies:** We provide a roadmap for diverse data curation, algorithmic fairness audits, and continuous monitoring to reduce disparities in AI-driven healthcare.**Interdisciplinary Policy Recommendations:** We offer actionable guidelines for policymakers, healthcare institutions, and AI developers, ensuring compliance with emerging regulations such as the EU AIA and Food and Drug Administration (FDA)’s Software as a Medical Device (SaMD) framework.

By synthesizing these elements, this study bridges the gap between high-level AI ethics and practical implementation, contributing to the development of trustworthy, equitable, and sustainable AI in healthcare.

## 2. The Evolving Ethical Landscape of AI in Healthcare

### 2.1. Privacy, Security, and Data Sovereignty

Ensuring privacy, safeguarding security, and maintaining data sovereignty are cornerstone challenges in the integration of AI into healthcare systems. AI models are fundamentally dependent on large-scale health datasets, and their effectiveness often increases as more information becomes available [35,47]. However, this reliance on extensive and sometimes granular patient data raises critical ethical and legal considerations, including the risk of unauthorized access, data breaches, and misuse by third parties. While principles of ethical AI have proliferated, high-level guidelines alone cannot guarantee ethical practice. The experience in medicine—with its established professional norms, fiduciary duties, and tested accountability mechanisms—demonstrates that ethical principles must be accompanied by enforceable and context-specific regulatory frameworks if they are to be truly effective [48,49].

In recent years, the healthcare sector has experienced significant data breaches, underscoring the urgent need for robust cybersecurity measures. One of the most significant breaches occurred in 2015 when Anthem Inc., one of the largest health insurance companies in the USA, suffered a cyberattack compromising the personal data of approximately 78.8 million individuals, including names, birthdates, and Social Security numbers [50]. Similarly, the 2021 ransomware attack on Ireland’s Health Service Executive disrupted nationwide healthcare services, forcing the shutdown of IT systems and delaying critical medical procedures [51]. These incidents highlight the vulnerabilities of healthcare AI systems to cyber threats and reinforce the necessity of stringent encryption, real-time threat detection, and proactive risk mitigation strategies.

Table 1 outlines key challenges in AI-driven healthcare, including data sensitivity, regulatory gaps, and technological risks. Even de-identified health data can be re-identified, raising privacy concerns [47]. The role of private entities adds complexity, as commercial interests may conflict with patient privacy [35,49]. Data sovereignty varies globally, creating compliance challenges. The General Data Protection Regulation (GDPR) enforces strict patient rights in the EU, while the Health Insurance Portability and Accountability Act (HIPAA) in the United States primarily regulates clinical entities, leaving gaps in non-traditional health data governance [35,52]. As a result, current safeguards such as the HIPAA in the United States have shown their limitations in the era of “*big data*”, where healthcare information increasingly resides outside traditional clinical settings [52,53]. Cross-border data transfers are further hindered by fragmented regulations, limiting AI scalability. Moreover, increasing reliance on cloud-based AI infrastructure introduces additional cybersecurity risks, as evidenced by past cyberattacks on electronic health records and healthcare databases [54]. Addressing these issues requires harmonized global policies and stronger enforcement to ensure secure, ethical, and scalable AI in healthcare.

#### Data Sovereignty, Regional Regulations, and Cross-Border Implications

Data sovereignty—the notion that data are subject to the laws and governance structures of the nation in which it is collected—is a crucial dimension of healthcare AI. Regulatory environments differ significantly across jurisdictions. In the EU, the GDPR grants patients stronger rights over their personal health data, mandating transparency, consent, and the right to be forgotten, which directly influences how AI systems can be designed, trained, and deployed [49,55]. In the United States, HIPAA focuses primarily on covered entities such as healthcare providers and insurers, leaving large segments of health-adjacent data beyond its scope [48,52]. This incongruity between jurisdictions can produce fragmented standards, creating compliance challenges for multinational collaborations, international clinical trials, and AI tools that require globally sourced training data [35,55]. For example, a cloud implementation strategy in Germany must carefully navigate the GDPR’s rigorous requirements for data anonymization and patient consent, while a similar initiative in the United States might face less stringent but still significant HIPAA-based obligations [48]. Such disparities complicate cross-border AI applications, particularly those aiming to leverage large and diverse patient datasets to improve model generalizability and mitigate biases. Addressing this tension could require harmonizing international policies, encouraging data-sharing agreements grounded in mutual trust and shared ethical principles, and fostering an environment where digital health innovation can flourish without jeopardizing individual privacy or national regulatory prerogatives [47,52,55].

### 2.2. Bias and Fairness in AI Algorithms

The growing use of AI in healthcare has underlined a critical yet often underappreciated challenge: the presence of biases that can negatively affect clinical decision-making and patient outcomes. These biases often stem from data that AI algorithms are trained on—data that may reflect entrenched racial, gender, socioeconomic, and other societal inequities [56,57,58]. When poorly addressed, these biases have the potential to amplify existing disparities in healthcare and exacerbate health inequities, leading to suboptimal diagnoses, treatment recommendations, and patient outcomes, particularly among marginalized or underrepresented groups [59,60,61].

#### 2.2.1. Examining Bias in Real-World AI Healthcare Applications

Several studies have documented racial and ethnic biases in AI-driven clinical decision-support tools. For instance, Seyyed-Kalantari et al. (2021) demonstrated that chest X-ray AI classifiers systematically and selectively underdiagnosed conditions in underrepresented patient populations—particularly minority women—raising serious ethical concerns about deploying these models in real-world clinical settings [62]. Similarly, AI models designed to support dermatological diagnoses have been trained predominantly on lighter skin tones, resulting in less accurate detection of skin conditions in patients with darker skin [63]. These inequities reflect underlying imbalances in training datasets, which often contain limited or poorly characterized data from diverse patient populations.

Beyond race and ethnicity, gender-based biases have also surfaced. For example, some AI algorithms fail to account for gender differences in disease presentation or symptomatology. This can lead to misdiagnoses or delayed treatment for conditions known to present differently in women than in men [56,57]. Socioeconomic disparities further compound these issues; patients in low-income regions may have limited access to healthcare services, generating sparser and less representative data that algorithms may interpret inaccurately [57,58]. Taken together, these findings highlight that AI systems can inadvertently entrench existing inequities unless carefully designed, validated, and monitored in diverse patient populations.

#### 2.2.2. Mitigating Bias Through Diverse Datasets and Algorithmic Audits

Addressing bias in AI-driven healthcare applications requires a comprehensive, multi-pronged approach that ensures models are developed, validated, and deployed fairly across diverse patient populations. First, developers and healthcare institutions should actively collaborate with diverse communities to ensure AI systems reflect real-world patient diversity. This might include forming partnerships with patient advocacy groups, community health organizations, and underrepresented populations to co-design data collection strategies and validate AI performance in clinical settings [58,59]. Second, targeted data curation initiatives should be prioritized. AI models must be trained on balanced datasets that represent a wide range of racial, ethnic, gender, and socioeconomic groups. Instituting standardized data collection protocols and verifying demographic completeness would help ensure that AI-driven recommendations do not disproportionately disadvantage certain populations [63].

Third, bias-aware model development techniques should be implemented. Developers can incorporate algorithmic fairness strategies, such as reweighting training samples, adversarial debiasing, and differential model tuning, to mitigate biases before deployment [56,57]. Additionally, embedding interpretability tools within AI models can help clinicians and regulators identify potential biases in real-time. Fourth, algorithmic audits, independent validations, and ongoing performance evaluations are essential. AI models should be continuously monitored and stress-tested across different demographic subpopulations to detect and correct emerging biases. Independent validations by external researchers or institutions can further ensure that model performance and fairness claims hold up under diverse, real-world conditions. Developers must conduct external fairness audits, report demographic-specific model performance, and implement bias mitigation updates based on real-world usage data [56,57].

Finally, policy-driven accountability mechanisms should incentivize fairness and transparency. Regulators and healthcare institutions should require AI companies to document bias mitigation efforts, disclose demographic performance metrics, and undergo independent third-party audits before clinical deployment. Ethical guidelines should mandate AI explainability reports, ensuring that healthcare providers understand how bias is detected and addressed in AI-generated decisions. By integrating community partnerships, targeted data curation, fairness-focused algorithm design, rigorous auditing, and policy-driven accountability, stakeholders can build more equitable, trustworthy AI models that improve healthcare outcomes for all populations rather than exacerbating existing disparities.

### 2.3. Regulatory and Legal Considerations

The rapid proliferation of AI technologies in healthcare—particularly large language models (LLM) and generative AI—has far outpaced traditional regulatory frameworks, exposing numerous gaps in oversight and governance. Current regulatory models, such as those from the U.S. FDA or the emerging EU AIA, were not originally designed to accommodate the complexities and iterative nature of AI systems, nor their reliance on evolving data sources and algorithms [23,64,65] (see Table 2 for a summary of key regulatory and legal considerations). Although these frameworks have begun to address medical AI devices, they often focus on initial performance evaluations rather than continuous monitoring, post-market surveillance, or the dynamic updates that characterize AI models.

#### 2.3.1. Limitations of Current Regulatory Frameworks

The FDA’s approval processes for medical AI devices, while thorough in traditional contexts, have demonstrated limitations when confronted with the more flexible, data-driven nature of AI [64]. Many approved AI tools undergo testing primarily at single clinical sites, a practice that can obscure how these models could perform in diverse healthcare settings, patient populations, or data-shift scenarios. The regulatory emphasis is often on static performance metrics at the time of approval rather than the ongoing validation required to ensure models remain robust and unbiased as real-world conditions evolve. Moreover, the U.S. FDA has struggled to establish a standardized, AI-specific regulatory framework, leading to uncertainties around continuous learning systems that adapt over time. Unlike traditional medical devices, AI models can change dynamically post-deployment, requiring new governance approaches that are currently underdeveloped in U.S. regulatory policy [66].

Similarly, the EU’s AIA takes a human-centric and risk-based approach, specifying categories of AI applications and requirements for “high-risk” systems [65]. Yet, while the AIA outlines processes for documentation, transparency, and risk assessment, there remain ambiguities regarding how AI systems could be monitored post-market, especially when updates to algorithms alter their risk classification. For example, an AI-driven diagnostic tool initially classified as low-risk could become high-risk as it incorporates real-world patient data, yet the current version of the AIA does not provide clear adaptive oversight mechanisms to address such transitions [67]. These gaps hinder AI adoption and create compliance burdens for healthcare providers, developers, and regulators.

In contrast, the United Kingdom (UK)’s NHS AI regulations provide a more flexible, iterative approach through the NHS AI Lab and National Institute for Health and Care Excellence frameworks. These models emphasize real-world evidence collection and phased AI validation but lack legally binding directives, making compliance voluntary rather than mandatory [68]. This regulatory inconsistency creates uncertainty for AI developers seeking global market approval, as they must navigate conflicting guidelines across different jurisdictions. As a result, healthcare AI innovations that comply with NHS frameworks may still face hurdles in meeting EU or United States standards, delaying patient access to new technologies [69].

Environmental implications and the underutilization of AI for sustainability—a dimension increasingly recognized as essential to achieving a “*good AI society*”—remain underemphasized in current EU regulatory efforts, highlighting the need for broader considerations beyond purely anthropocentric viewpoints [65]. Additionally, sectors such as AI-driven mental health diagnostics, digital therapeutics, and patient-facing AI chatbots remain largely unregulated, exposing vulnerabilities in accountability, liability, and patient safety [70].

#### 2.3.2. Adapting to the Rapid Evolution of AI Technologies

In an AI ecosystem marked by rapid innovation and frequent model iteration, static, one-time approval processes are insufficient. LLMs and generative AI systems can learn, adapt, and even generate synthetic data, complicating the validation processes and risk assessments that traditional regulatory frameworks rely upon [23,71,72,73]. To keep pace, regulators must shift towards more adaptive, iterative, and evidence-based oversight models that emphasize continuous monitoring, real-time auditing, and dynamic compliance checks. For example, successful implementation of the sustainable EU AIA requires interdisciplinary efforts in real-life applications [74,75].

Several approaches can help achieve this adaptation. First, governance models that mandate periodic re-certification or ongoing performance evaluations after deployment can ensure that AI systems maintain their safety, accuracy, and fairness [71,73]. Second, cooperative approaches involving multiple stakeholders—healthcare providers, AI developers, patient advocacy groups, ethicists, and policymakers—can yield more responsive regulatory regimes that anticipate potential issues rather than merely reacting to them [23,73]. Such adaptive frameworks could incorporate post-market surveillance, algorithmic impact assessments, and mechanisms for rapid recalls or adjustments when models degrade or exhibit biases.

Additionally, frameworks such as AI4People’s ethical guidelines and governance recommendations provide templates for integrating ethical principles directly into regulatory structures, ensuring that notions of transparency, accountability, fairness, and patient autonomy are not just aspirational ideals but enforceable standards [71,73]. Encouraging the adoption of sustainability and equity goals as part of regulatory mandates can help ensure that advancements in healthcare AI align with broader societal interests, including reduced healthcare disparities and environmental stewardship [65,71].

**Table 2 jcm-14-01605-t002:** Key regulatory and legal considerations for AI in healthcare.

Aspect	Key Challenges	Current Frameworks	Limitations	Recommendations	Reference(s)
FDA Approval Processes	Evaluating AI devices pre-market with static metrics	FDA Guidance on Software as a Medical Device and AI tools	Often rely on single-site testing, insufficient generalizability, limited post-market surveillance	Implement iterative approvals, continuous performance monitoring, and multi-site validation	He et al., 2019 [23]; Wu et al., 2021 [64]
EU Artificial Intelligence Act (AIA)	Establishing a risk-based, human-centric approach	EU AIA Proposal	Lacks adaptation to evolving AI models and overlooks environmental and underuse aspects	Develop adaptive standards, include environmental and sustainability considerations, and broaden the scope beyond a human-centric focus	Pagallo et al., 2022 [65]; Montag & Finck, 2024 [74]
Ethical and Governance Frameworks	Ensuring fairness, accountability, and transparency	AI4People’s ethical guidelines, EU and national ethics boards	Many guidelines remain aspirational, not fully integrated into enforceable legal structures	Integrate ethical principles into legal mandates, adopt periodic audits, and enforce transparency and bias reporting	Floridi et al., 2018 [71]; Reddy et al., 2020 [72]
Sustainability and Equity in Regulation	Addressing underutilization of AI for environmental or equitable health outcomes	Limited or absent in current regulations	Insufficient emphasis on sustainability, non-financial disclosures, and equitable data use	Require eco-impact assessments, enforce duties of care, promote equitable access and representativeness in datasets	Palkova, 2021 [73]; Pagallo et al., 2022 [65]; Hacker, 2024 [75]
Continuous Adaptation to Rapid Evolution	Accounting for model updates, emergent behaviors of LLMs, and generative AI	Static approval models, limited version control	Regulatory gaps for dynamic updates, limited real-time oversight of algorithmic changes	Implement ongoing validation, create rapid response mechanisms, encourage multi-stakeholder regulatory development	He et al., 2019 [23]; Reddy et al., 2020 [72]; Palkova, 2021 [73]

## 3. From Principles to Practice: Defining and Evaluating Trustworthy AI

As AI’s role in healthcare expands, moving from high-level ethical principles to concrete implementation strategies becomes an urgent priority. Robust frameworks that ensure transparency, explainability, and accountability are essential for cultivating trust in AI-driven medical systems. Beyond outlining abstract ideals, the healthcare industry and regulators must identify verifiable metrics and practical techniques to assess whether AI models operate fairly, accurately, and safely in clinical environments.

AI systems vary significantly in their ability to support trustworthy implementation in healthcare. Symbolic AI and rule-based expert systems provide greater explainability but often struggle with complexity in healthcare [76]. ML models, particularly deep learning architectures, offer superior predictive power but suffer from interpretability challenges, making them less transparent [77]. Meanwhile, hybrid AI approaches that integrate symbolic reasoning with deep learning have gained attention for their ability to balance transparency and predictive performance, for example, for targeted metabolomics analysis of diabetic retinopathy [78]. Identifying the most suitable AI category for specific clinical tasks is a crucial step toward ensuring trust and accountability in healthcare AI applications.

Figure 2 provides a conceptual framework for achieving Trustworthy AI by balancing helpful and harmful factors that influence AI integration and its impact on outcomes. The framework is structured along two axes:*Ease of Integration (horizontal axis)*—This axis represents the balance between helpful internal attributes that improve AI adoption and harmful external factors that create barriers, such as regulatory gaps and privacy risks. AI systems that emphasize fairness, explainability, and ethical governance tend to integrate more smoothly, while those facing compliance challenges or regulatory misalignment may struggle with adoption.*Impact on Outcomes (vertical axis)*—This axis differentiates internal organizational efforts (bottom) from external regulatory, ethical, and societal constraints (top). A higher impact on outcomes is observed when AI systems align with compliance and security needs, whereas a lower impact may arise from transparency and explainability issues that require internal efforts to improve AI adoption.

Each quadrant represents a critical factor influencing Trustworthy AI development:**Top-left (Fair, Accountable, and Bias-Free AI):** Highlights fairness, accountability, and equity in AI design, ensuring responsible decision-making and reducing biases in healthcare AI models.**Top-right (Secure, Compliant, and Privacy-Focused AI):** Represents the importance of regulatory adherence and cybersecurity measures, ensuring compliance but also presenting challenges if excessive constraints hinder innovation.**Bottom-left (Transparent, Reliable, and Explainable AI):** Focuses on AI systems that prioritize transparency and interpretability, enabling healthcare professionals to understand and trust AI-generated decisions.**Bottom-right (Sustainable, Scalable, and Trustworthy AI):** Emphasizes long-term viability, ethical scalability, and the alignment of AI deployment with sustainability and equity goals.

Applying this framework to specific AI categories can help stakeholders assess which models best fit different healthcare applications. For example, while deep learning may be more effective for complex imaging tasks such as tumor detection, rule-based systems might be preferable for clinical decision support where interpretability is paramount [79]. By aligning AI selection with clinical context, regulatory requirements, and ethical imperatives, healthcare organizations can ensure AI deployment is both effective and trustworthy [10].

This framework also allows stakeholders to assess their AI implementations by identifying strengths and areas for improvement. An AI model that excels in privacy compliance but lacks explainability would fall in the top-right quadrant and may need refinement to improve transparency. Likewise, healthcare institutions should shift from fragmented technical solutions toward a more holistic approach that integrates fairness, accountability, and sustainability. Achieving *Trustworthy AI* requires a continuous process of refinement, addressing regulatory gaps, and reinforcing ethical, transparent, and scalable AI practices that benefit both healthcare providers and patients.

### 3.1. Transparency and Explainability: Moving Beyond the Black Box

Ensuring trustworthy AI in healthcare requires not only fairness, accountability, and privacy but also transparency and explainability. One of the major barriers to trust is the “*black-box*” problem, where AI models—especially deep learning systems—lack interpretability, making it difficult for clinicians to understand their decision-making processes. This opacity can erode trust, pose safety risks, and hinder adoption in clinical practice [80,81,82,83]. To effectively enhance transparency, stakeholders must implement clear metrics and techniques that improve AI interpretability. Our analysis systematically selected empirical studies, regulatory frameworks, and industry guidelines from PubMed, Scopus, and Web of Science to evaluate the effectiveness of existing explainability methods in healthcare AI. Referring back to the conceptual framework shown in Figure 2, transparency and explainability are essential internal attributes that move AI systems closer to trustworthiness. AI developers should focus on interpretable model architectures, post-hoc explainability techniques, and user-friendly decision-support tools that bridge the gap between complex algorithms and clinical applicability. When organizations prioritize transparency initiatives, they move closer to the center of Figure 2, balancing trust-building attributes against the challenges of real-world integration.

#### 3.1.1. Proposing Measurable Metrics for Transparency in AI Systems

To advance beyond abstract principles, measurable indicators of transparency must be identified. Such metrics might include the percentage of model outputs accompanied by interpretable explanations, the level of detail these explanations provide, or standardized “*explainability scores*” that gauge how easily a human can comprehend the reasoning behind an AI’s decision [16,83,84]. Implementing explainability scores involves several approaches:**Feature Contribution Analysis:** Techniques such as SHapley Additive exPlanations (SHAP) assign an importance value to each input feature, indicating its contribution to the model’s output [85]. For instance, in a clinical decision support system predicting patient outcomes, SHAP values can highlight which clinical parameters (e.g.*,* blood pressure, cholesterol levels) most significantly influenced a specific prediction. This allows healthcare professionals to understand the model’s reasoning and assess its alignment with clinical knowledge [11].**Local Interpretable Model-agnostic Explanations (LIME):** LIME approximates the AI model locally with an interpretable model to explain individual predictions [86]. In practice, if an AI system recommends a particular treatment plan, LIME can provide a simplified explanation of that specific recommendation, enabling clinicians to evaluate its validity before implementation [11].**Attention Mechanisms in Neural Networks:** In models such as attention-based Bidirectional Long Short-Term Memory networks, attention weights can serve as proxies for feature importance, offering insights into which aspects of these input data the model focuses on during prediction. For example, in patient monitoring systems, attention mechanisms can help identify critical time periods or vital signs that the model deems most relevant, thereby providing transparency in its decision-making process [87].

In clinical contexts, these could be mapped to real-world performance measures—such as the rate at which physicians can accurately identify a model’s reasoning errors or the time it takes for healthcare staff to understand and trust an algorithm’s recommended treatment option [23]. Setting these measurable goals encourages iterative improvement, aligning with the idea of AI as an evolving ecosystem that thrives on continuous feedback and regulatory oversight. In terms of the framework in Figure 2, establishing metrics for transparency can help healthcare organizations move from a purely aspirational stance to a more outcomes-focused approach. By systematically assessing transparency, they ensure that this dimension of AI trustworthiness is not just a guiding principle but a quantifiable target.

Another promising approach involves developing structured documentation protocols—akin to model “nutrition labels”—that summarize the model’s data sources, training methodology, known limitations, and prior performance across diverse patient populations [82]. By employing standardized reporting frameworks, stakeholders can more consistently evaluate the transparency and trustworthiness of AI implementations, thereby reinforcing the beneficial factors that drive the healthcare ecosystem toward trustworthy AI.

#### 3.1.2. Advancements in Explainability Techniques

In parallel, the technical community has made considerable progress in rendering complex models more comprehensible. Model-agnostic techniques such as LIME and SHAP can be applied to a wide range of black-box models, highlighting the most influential features that drive particular predictions [80,88]. These methods help clinicians understand why an AI tool suggests a particular diagnosis or treatment, enabling them to identify potential misjudgments, ensure consistency with medical knowledge, and communicate the rationale to patients.

Other efforts focus on building transparency into the model design itself. Intrinsically interpretable models—such as decision trees or rule-based systems—prioritize understandability from the outset, albeit sometimes at the expense of accuracy. Hybrid solutions seek to bridge this gap by pairing high-performing black-box models with interpretable surrogate models or post-hoc explanation layers that maintain performance while offering human-readable logic [71,82]. Neural networks, historically challenging to interpret, now benefit from visualization methods and layered explanations that reveal which inputs strongly influence a network’s decisions [89]. These visualizations, especially for text-based models, can be integrated into clinical workflows, allowing clinicians to trace how different textual inputs—such as patient history or lab values—affect model outputs [90,91].

#### 3.1.3. Building Trust Through Understanding and Shared Accountability

Combining robust explainability techniques with clear performance metrics not only aids healthcare practitioners in making informed decisions but also supports regulators in assessing compliance with emerging guidelines that call for AI transparency. From the perspective of “*towards regulatory genomes*”, scalable explainability standards ensure that trust-building measures are not ad hoc but systematically embedded into the lifecycle of AI development and deployment [92]. Ultimately, as AI systems evolve, their transparency and comprehensibility must evolve in tandem, ensuring that high-stakes healthcare decisions are guided by algorithms that can justify their reasoning, adapt to diverse patient populations, and maintain trust across the entire healthcare ecosystem.

In selecting references for this section, we provided a balance between technical solutions, regulatory considerations, and case studies. This approach allowed us to provide a holistic perspective on how transparency can be operationalized in healthcare AI. As illustrated in Figure 2, every step—establishing measurable transparency goals, adopting cutting-edge explainability methods, and enforcing accountability—guides healthcare AI closer to the central goal of achieving *Trustworthy AI*. In this process, internal organizational efforts bolster reliability and fairness while managing potentially harmful external influences such as regulatory gaps or privacy risks. The result is a more resilient, ethically grounded, and equitable healthcare AI ecosystem, capable of adapting its transparency and comprehensibility to diverse patient populations and changing clinical contexts.

### 3.2. Accountability in AI Decision-Making

As AI systems increasingly influence clinical judgments and patient outcomes, the need for clear accountability frameworks becomes paramount. Unlike traditional medical tools, AI-driven diagnostic and therapeutic recommendations can arise from complex, opaque models that challenge existing legal and ethical paradigms. Who bears responsibility if an autonomous surgical robot makes an error or if an ML algorithm systematically disadvantages a particular patient population? These questions demand rigorous standards, robust governance models, and transparent lines of accountability to maintain trust and integrity in healthcare AI.

In the domain of autonomous robotic surgery, for example, a structured approach to accountability must distinguish between accountability, liability, and culpability (see Table 3). While “*accountability*” relates to the obligation of an entity—be it a developer, healthcare provider, or institution—to answer for the performance of AI-driven tools, “*liability*” involves potential legal consequences and compensation for the harm caused. “*Culpability*” addresses moral wrongdoing, which is challenging to apply to non-human agents. O’Sullivan and colleagues highlight how future surgical robots capable of performing routine tasks under human supervision could require new frameworks to apportion responsibility and uphold patient safety [93]. In this scenario, the concept of a “doctor-in-the-loop” ensures that clinicians retain ultimate control and ethical oversight, bridging current clinical responsibilities with the emerging capabilities of autonomous systems.

Beyond clinical interventions, accountability in AI healthcare systems also depends on embedding ethical principles into practical governance structures. As Shneiderman suggests, multi-level strategies are essential for ensuring reliable, safe, and trustworthy AI [94]. At the team level, adherence to rigorous software engineering practices—such as documentation, traceability, and thorough verification—can facilitate post-hoc analyses of failures. At the organizational level, a strong safety culture promotes continuous improvement and learning from errors. At the industry and societal level, external audits, regulatory oversight, and professional standards can provide independent checks on the trustworthiness of AI. Together, these measures help clarify who is accountable when AI models perform poorly, exacerbate disparities, or cause patient harm.

From a broader policy perspective, Morley and colleagues outline how accountability is integral to ethical AI in healthcare, linking it to issues of epistemic soundness, fairness, and system-level governance [95]. Ethical guidelines and regulatory principles must align with clinical realities, ensuring that decision-makers, regulators, and developers remain answerable for their contributions. Formalizing accountability encourages ongoing stakeholder engagement—patients, clinicians, policymakers, and developers—thereby strengthening public trust and reducing the risk of a loss of confidence that could hinder AI adoption in healthcare. Table 3 provides a structured overview of the key dimensions of accountability in healthcare AI, outlining critical components and their implications.

## 4. Future Directions: A Roadmap for Trustworthy AI in Healthcare

As AI becomes increasingly integral to healthcare, meeting the demand for trustworthy, equitable, and patient-centered solutions calls for a forward-looking, strategic approach. The future depends on cultivating strong collaborations among developers, clinicians, policymakers, and ethicists, as well as fostering global standards and adaptive regulatory models. Emphasis must shift from reactive measures to proactive frameworks that anticipate emerging challenges, promote sustainability, and ensure that AI-driven care remains both ethically grounded and clinically valuable.

A key challenge is the lack of quantifiable frameworks for assessing AI governance. To address this, we propose a structured approach that integrates clear performance indicators, policy recommendations, and real-world implementation strategies. This future-oriented perspective recognizes that current ethical, regulatory, and infrastructural approaches must evolve in tandem with the technologies they govern. Transparent performance metrics, bias mitigation strategies, and robust oversight mechanisms must become the norm. Moreover, we must expand our understanding of AI’s long-term implications—both environmental and socioeconomic—ensuring that breakthroughs in precision medicine do not come at the expense of sustainability, inclusivity, or public trust. Table 4 highlights a forward-looking, holistic approach to AI governance in healthcare.

### 4.1. Cross-Disciplinary Collaboration for AI Governance

Ensuring that AI systems are both clinically effective and ethically sound requires comprehensive governance frameworks developed through cross-disciplinary collaboration. This study adopts an interdisciplinary perspective, drawing upon AI ethics, legal frameworks, clinical practice, and regulatory science to propose actionable recommendations. For instance, regulatory adaptation strategies must incorporate insights from medical professionals to ensure that AI compliance mechanisms do not hinder clinical efficiency, while ethicists and technologists contribute to the formulation of fairness and bias mitigation techniques. AI developers, healthcare practitioners, policymakers, and ethicists each bring unique perspectives and expertise to the table, shaping AI policies and standards that reflect the realities of clinical care, technological limitations, societal values, and regulatory imperatives [96,97,98]. Such cross-disciplinary dialogue is essential for anticipating potential ethical dilemmas, addressing biases, and ensuring that AI tools enhance patient care without inadvertently harming underserved communities.

The complexity of healthcare settings, coupled with the rapid evolution of AI technologies, means that no single stakeholder can grasp all the implications. Policymakers and public sector administrators provide insights into regulatory structures and enforcement mechanisms that maintain accountability and fairness. Healthcare professionals contribute to the clinical context, ensuring that AI-driven solutions align with patient needs and integrate seamlessly into existing workflows. Ethicists help ensure that principles such as fairness, autonomy, and transparency are not only articulated but operationalized. AI developers, for their part, translate these principles into technical architectures, develop interpretable models, and maintain robust cybersecurity measures [96,99,100]. By involving all these groups from the outset, a “*design for governance*” approach becomes more feasible, and the resulting AI frameworks become more pragmatic, inclusive, and sustainable.

International collaborations play a crucial role in developing global AI standards and aligning regulatory frameworks across jurisdictions. Key organizations such as the International Organization for Standardization (ISO) and the International Electrotechnical Commission (IEC) have jointly introduced ISO/IEC 42001:2023 [101], the first global AI management system standard, providing a structured approach for responsible AI deployment. Similarly, the IEEE Standards Association has established IEEE 2801-2022 (IEEE Recommended Practice for the Quality Management of Datasets for Medical Artificial Intelligence), a framework for ensuring quality management of datasets used in AI-driven medical devices [102]. These initiatives set international benchmarks that help standardize AI governance, ensuring transparency, fairness, and accountability across different regions.

Beyond technical standards, industry and government collaborations are essential for establishing consistent ethical guidelines and bridging regulatory differences [103,104]. For instance, the SPIRIT-AI and CONSORT-AI guidelines provide international standards for clinical trials of AI systems, ensuring complete and transparent reporting of clinical trial protocols and reports involving AI interventions [105]. Additionally, the FUTURE-AI consortium has developed an international consensus guideline for trustworthy and deployable artificial intelligence in healthcare, addressing technical, clinical, ethical, and legal considerations [106].

Industry and government collaborations are also essential for establishing consistent ethical guidelines and bridging regulatory differences. While government agencies provide baseline policies, global consortia, professional organizations, and private sector alliances refine these into practical, scalable governance models. These partnerships facilitate knowledge exchange, accelerate policy development, and ensure AI regulations align with the United Nations SDGs. As AI becomes more embedded in healthcare, governance must evolve just as rapidly. Effective AI governance requires an interdisciplinary approach, bringing together technologists, clinicians, policymakers, and ethicists to create a healthcare AI ecosystem that is reliable, equitable, and patient-centered [107]. By fostering international collaboration, stakeholders can drive responsible AI adoption, ensuring healthcare AI solutions are ethically sound, technically robust, and globally harmonized.

### 4.2. Integrating AI with Ethical Frameworks: Towards “Ethical by Design” AI

Building trust in healthcare AI requires more than after-the-fact adjustments; it calls for weaving ethical considerations directly into the fabric of AI systems from the start. An “*Ethical by Design*” approach ensures that core principles—such as fairness, safety, privacy, and accountability—are not retrofitted but form the foundation of an AI system’s architecture, algorithms, and operational protocols [108,109,110]. Rather than viewing ethics as an external checkpoint, developers, clinicians, and policymakers should treat it as a guiding compass that informs technical decisions, clinical integration, and policy formation.

In this model, developers collaborate closely with ethicists, legal experts, and healthcare professionals to anticipate potential harm or biases early in the design process. Techniques from responsible AI frameworks, such as continuous impact assessments, formalized documentation, and user-focused explainability tools, support proactive decision-making [93,108,109]. For example, a surgical robot could be engineered with clearly defined accountability layers and operational constraints that safeguard patients, ensuring a human surgeon remains “in the loop” for critical decisions [93]. Similarly, AI-driven diagnostic tools can be designed with embedded fairness checks and transparency features, enabling clinicians and patients to understand how predictions are made and to identify instances of bias or error.

Operationalizing “*Ethical by Design*” also entails going beyond surface-level compliance. It involves implementing iterative audits, rigorous validation protocols, and clear escalation paths for addressing misconduct or errors. Ethical frameworks become living documents that adapt to evolving technology and data landscapes, thus ensuring that AI systems remain aligned with core human values as they advance [111,112]. Thus, this integrated approach not only strengthens trust and acceptance of AI in healthcare but also enhances clinical outcomes by making ethical integrity, patient well-being, and responsible innovation inseparable from the development and deployment lifecycle.

### 4.3. Sustainable and Ethical AI Development

The pursuit of trustworthy AI in healthcare requires a broader perspective that considers not only the immediate clinical benefits of AI systems but also their long-term environmental, social, and economic implications. Sustainable AI extends beyond designing algorithms for clinical accuracy and efficiency; it requires a lifecycle approach—from data sourcing and model training to deployment and decommissioning—according to principles of ecological integrity, social justice, and intergenerational equity [46,113]. In short, sustainable AI must both support global sustainability goals and operate in an environmentally responsible manner.

One pressing concern is the substantial energy consumption and environmental footprint associated with training and maintaining large AI models, often powered by resource-intensive data centers [114,115]. Healthcare organizations, already under scrutiny for their contribution to greenhouse gas emissions and environmental stressors, must weigh the ecological costs of their AI investments. Striving for “green AI” involves:**Optimizing algorithms for energy efficiency**, reducing computational complexity while maintaining performance.**Leveraging renewable energy** to power AI infrastructure, such as cloud providers committed to carbon neutrality.**Implementing hardware recycling strategies** to reduce e-waste from outdated computing resources [116].

To assess the environmental impact of AI systems, several tools and metrics have been developed:**Carbontracker**: This tool monitors hardware power consumption and local energy carbon intensity during the training of deep learning models, providing accurate measurements and predictions of the operational carbon footprint [117].**Energy Usage Reports**: This approach emphasizes environmental awareness as part of algorithmic accountability, proposing the inclusion of energy usage reports in standard algorithmic practices to promote responsible computing in ML [118].

By utilizing these tools, stakeholders can effectively measure and mitigate the environmental burden of AI systems. Beyond environmental sustainability, ethical sustainability requires that AI promotes equitable access to healthcare resources and does not exacerbate existing disparities [24,113]. AI tools must be designed and validated to avoid amplifying biases or producing harmful downstream effects, such as misinformation or discriminatory treatment paths. As AI applications increasingly influence healthcare policies and global health initiatives, close alignment with the United Nations SDGs ensures that AI-driven innovations address societal needs without compromising future generations’ ability to meet their own [24,35]. Therefore, sustainable AI development calls for a multi-stakeholder approach:**Policymakers** should establish incentives and regulations that promote responsible innovation.**Healthcare institutions** should adopt procurement policies favoring energy-efficient AI solutions and transparent supply chains.**AI developers** should integrate sustainability metrics into model evaluation frameworks, ensuring that AI systems align with both ethical and environmental standards.

By combining environmental stewardship, ethical rigor, and a commitment to the long-term well-being of communities, healthcare stakeholders can ensure that AI not only enhances patient outcomes today but also preserves the ecological and moral foundations upon which future healthcare depends.

### 4.4. Establishing Quantifiable AI Governance Metrics

To bridge the gap between high-level ethical principles and practical implementation, we introduce a set of quantifiable governance metrics that evaluate AI systems based on transparency, bias mitigation, regulatory compliance, and sustainability. These dimensions can be measured using specific performance metrics outlined in Table 5.

These quantifiable measures can allow AI stakeholders to systematically assess the effectiveness and ethical alignment of AI systems in healthcare. By applying this framework, organizations could track AI performance over time, improve regulatory compliance, and ensure patient safety.

### 4.5. Policy Recommendations for AI Governance Implementation

To operationalize these metrics, we propose the following policy-driven interventions for healthcare AI governance:**Mandatory Transparency Reporting**: AI developers must publish detailed model documentation, including interpretability techniques, demographic performance disparities, and post-market monitoring results. For example, the UK NHS AI Lab mandates model transparency reports for all approved AI-driven clinical tools [68].**Bias Mitigation Mandates**: Regulators should require AI developers to conduct fairness audits and actively mitigate algorithmic biases before deployment. For example, the FDA’s SaMD regulatory framework includes demographic fairness testing in clinical AI validation [66].**Real-Time AI Performance Audits**: An AI Governance Task Force should oversee post-market AI performance, ensuring continuous validation and bias correction. For example, the EU AIA proposes AI sandboxes for continuous evaluation and refinement [74].**Sustainability Compliance Incentives**: AI developers adhering to energy efficiency standards should receive tax incentives or expedited regulatory approvals. Initiatives supporting the development and implementation of AI solutions that contribute to carbon neutrality can offer subsidies to organizations integrating sustainable AI technologies, thereby reducing their carbon footprint. Studies have shown that AI-based solutions can significantly reduce energy consumption and carbon emissions in various sectors [116,117,118].

These policy interventions ensure governments and healthcare institutions enforce AI governance metrics effectively, bridging the gap between theoretical AI ethics and real-world implementation.

## 5. Conclusions

The integration of AI into healthcare presents both remarkable opportunities and profound responsibilities. While AI enhances diagnostic precision, treatment efficacy, and patient access, its ethical, regulatory, and societal challenges require careful governance. This study contributes by providing a structured, interdisciplinary framework that integrates AI ethics, policy, and deployment strategies, ensuring transparency, fairness, accountability, and sustainability. We introduce quantifiable trustworthiness metrics, analyze AI categories for clinical suitability, and propose the *Regulatory Genome*—an adaptive framework aligning AI governance with global regulatory trends and SDGs. Additionally, we outline bias mitigation strategies and offer policy recommendations to support compliance with emerging regulations such as the EU AI Act and the FDA’s SaMD framework. By synthesizing insights from healthcare, ethics, and technology policy, this study offers a comprehensive approach to responsible AI deployment, ensuring its equitable and sustainable integration into healthcare systems. To strengthen AI governance and ensure its ethical implementation, we propose the following actionable recommendations:**Regulatory Adaptation**—Policymakers should develop flexible, adaptive regulations that evolve alongside AI technologies, ensuring continuous monitoring, transparency reporting, and risk assessment frameworks.**Interdisciplinary Cooperation**—Collaboration among technologists, clinicians, ethicists, and policymakers is essential to establish standardized guidelines that balance innovation with ethical safeguards.**Global Policy Alignment**—International cooperation should aim to harmonize AI regulations across jurisdictions, addressing disparities in privacy laws, data sovereignty, and compliance standards.**Research Priorities**—Future studies should focus on empirical evaluations of AI fairness, real-world implementation of AI transparency frameworks, and the development of sustainability metrics for AI-driven healthcare.

By addressing governance gaps and proposing actionable strategies, this study contributes to ongoing discussions on ethical AI in healthcare. Moving forward, collaborative efforts between academia, industry, and regulatory bodies could be critical to maximizing AI’s benefits while mitigating risks and ensuring equitable, sustainable AI adoption.

## Figures and Tables

**Figure 1 jcm-14-01605-f001:**
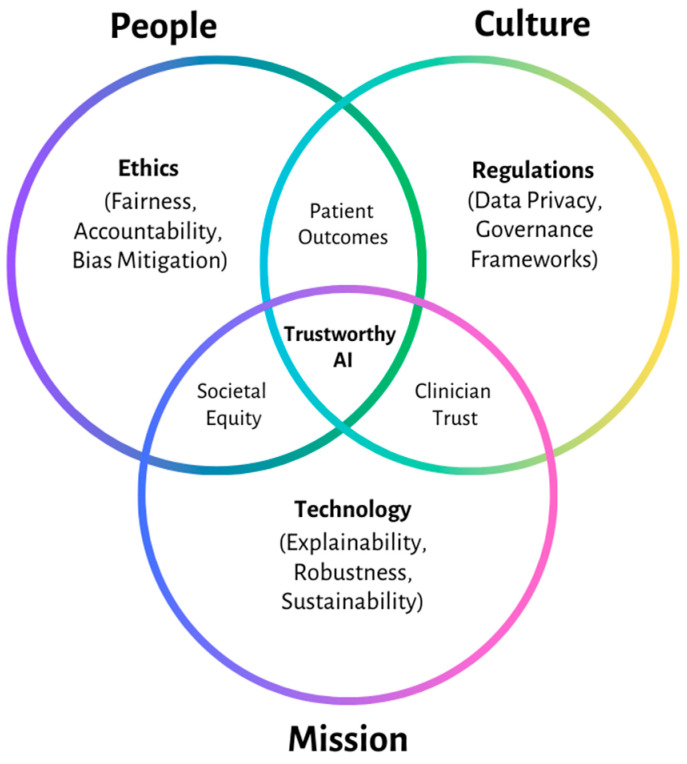
Key components of trustworthy AI in healthcare: The intersection of ethics, regulations, and technology, emphasizing privacy, fairness, and accountability for ethical AI integration.

**Figure 2 jcm-14-01605-f002:**
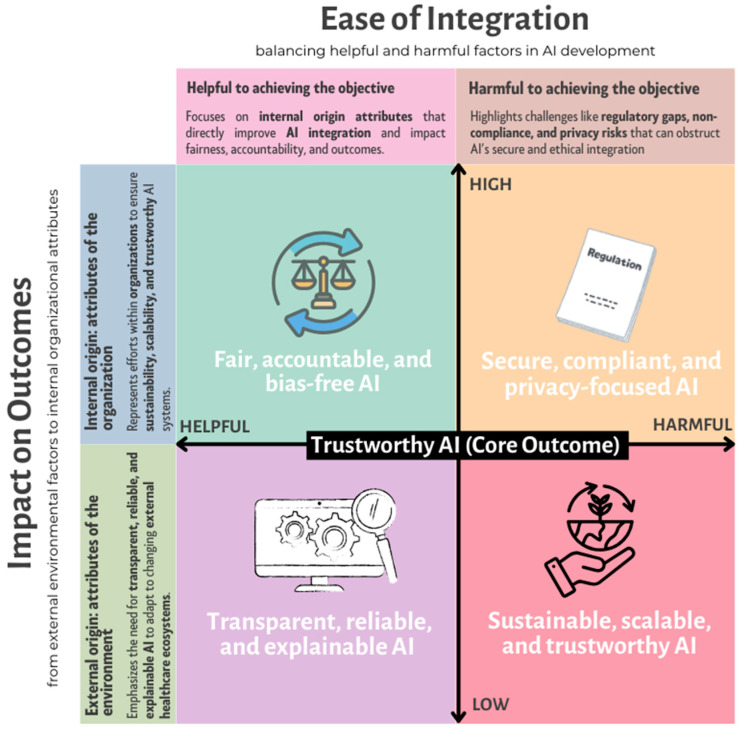
Conceptual framework illustrating how helpful internal attributes and challenging external factors intersect to influence AI outcomes. Moving toward the center signifies progress in achieving “*Trustworthy AI*” by balancing dimensions such as fairness, accountability, transparency, compliance, privacy, and sustainability. Each quadrant represents a key facet—ranging from bias-free fairness to secure privacy measures—that, when integrated and optimized, leads to safer, more effective, and ethically sound AI systems in healthcare.

**Table 1 jcm-14-01605-t001:** Key aspects of privacy, security, and data sovereignty in AI-driven healthcare.

Aspect	Description	Example/Implications	Reference(s)
Sensitive Data Handling	AI systems depend on vast datasets, creating risks of breaches, re-identification, and privacy violations	- Advanced algorithms may re-identify anonymized data. - Patient data transfer across institutions increases vulnerabilities	Perni et al., 2023 [47]
Private Entities	Private organizations managing health data may prioritize competing goals over patient privacy	- Public-private partnerships with poor oversight may weaken safeguards	Mittelstadt, 2019 [35]; Murdoch, 2021 [49]
Data Sovereignty	Governance of patient data varies across regions, creating challenges for multinational applications of AI in healthcare	- GDPR enforces stringent privacy rules in the EU. - HIPAA covers only clinical entities in the US, leaving gaps in coverage	Cohen & Mello, 2018 [52]; Mittelstadt, 2019 [35]; Murdoch, 2021 [49]; Imrie et al., 2023 [48]; Putzier et al., 2024 [55]
Cross-Border Challenges	Fragmented global regulations complicate AI training and deployment on diverse datasets	- Differences in GDPR and HIPAA hinder cross-border data sharing	Mittelstadt, 2019 [35]; Imrie et al., 2023 [48]; Putzier et al., 2024 [55]
Regulatory Gaps	Existing frameworks such as HIPAA are outdated for addressing AI’s demands in healthcare, particularly for big data applications	- HIPAA creates delays and costs for researchers without consistently enhancing privacy	Ness, 2007 [53]; Murdoch, 2021 [49]
Technological Risks	Cloud computing and digital health solutions offer scalability but introduce potential vulnerabilities to cyberattacks	- German healthcare faces GDPR compliance issues with cloud integration	Imrie et al., 2023 [48]; Putzier et al., 2024 [55]

**Table 3 jcm-14-01605-t003:** Key dimensions of accountability in healthcare AI.

Aspect	Description	Examples/Implications	Reference(s)
Accountability vs. Liability	Accountability refers to the obligation to explain and justify actions; liability involves legal responsibility and potential financial reparations	Healthcare providers, AI developers, and institutions must understand their roles. If an autonomous surgical robot makes errors, who is answerable, and who bears legal blame?	O’Sullivan et al. 2019 [93]
Culpability	Pertains to moral wrongdoing, challenging to assign to non-human agents	AI systems lack moral agency, complicating culpability. The emphasis shifts to human oversight, ensuring “*doctor-in-the-loop*” models of surgical AI	O’Sullivan et al. 2019 [93]
Multi-Level Governance	Accountability structures span team, organizational, industry, and regulatory levels	At the team level: rigorous documentation and audits. At the organizational level: safety culture and continuous improvement. At industry/regulatory level: standards, certifications, and external oversight	Shneiderman, 2020 [94]; Morley et al., 2020 [95]
Embedding Ethical Principles	Aligning AI design and deployment with ethical norms to ensure fairness, transparency, and responsibility	Formalized guidelines and professional codes help clarify stakeholder duties, reduce disparities, and maintain public trust	Shneiderman, 2020 [94]; Morley et al., 2020 [95]
Stakeholder Engagement	Inclusive engagement of clinicians, patients, developers, policymakers, and regulators	Regular input from diverse stakeholders ensures that accountability frameworks reflect real-world clinical needs and societal values, maintaining trust and legitimacy	Shneiderman, 2020 [94]; Morley et al., 2020 [95]

**Table 4 jcm-14-01605-t004:** Strategic roadmap for Trustworthy, Ethical, and Sustainable AI in healthcare.

Priority Area	Key Challenges	Proposed Actions	Primary Stakeholders	Intended Impact
Cross-Disciplinary Collaboration	Fragmented expertise, misaligned incentives, and limited communication across AI developers, clinicians, policymakers, and ethicists	Establish multidisciplinary working groups, encourage joint training programs, and foster international consortia for standards-setting and knowledge exchange	AI developers, healthcare practitioners, policymakers, ethicists, industry consortia, professional associations	More cohesive governance frameworks, improved policy relevance, enhanced credibility and trust in AI solutions
Ethical by Design Integration	Retrofitted ethics checks, delayed detection of biases, and insufficient accountability mechanisms embedded in technology	Embed ethical principles (fairness, accountability, privacy) at the inception of model development, implement continuous impact assessments, adopt formalized “*ethical-by-design*” guidelines and iterative audits	AI developers, ethicists, regulators, legal experts, clinical oversight boards	Earlier identification of risks, stronger patient protections, higher adoption rates due to enhanced trust and transparency
Sustainable and Equitable AI	High energy consumption, environmental degradation, unequal access to advanced AI tools, and risk of exacerbating health disparities	Optimize algorithms for energy efficiency, source renewable computing power, enforce procurement policies prioritizing sustainable vendors, integrate fairness checks, and align with global sustainability goals	Healthcare institutions, AI developers, environmental bodies, public health officials, sustainability experts	Reduced ecological footprint, long-term resource stewardship, equitable access to advanced healthcare tools, alignment with Sustainable Development Goals
Adaptive Oversight and Regulation	Static, fragmented regulatory models that cannot keep pace with rapidly evolving AI technologies	Develop adaptive regulatory frameworks, promote iterative certification models, support real-time performance monitoring, encourage flexible policy experimentation	Policymakers, regulatory agencies, professional standards organizations, industry partners	Responsive oversight that evolves with emerging technologies, ensuring continuous compliance, safety, and patient well-being
Public Engagement and Transparency	Low patient and public trust due to opaque decision-making, fear of unchecked innovation, and limited patient input	Provide accessible explanations of AI decisions, facilitate patient and community forums, public reporting of model performance and bias audits, and integrate user feedback into ongoing improvements	Patient advocacy groups, healthcare institutions, policymakers, AI developers, civil society organizations	Heightened public trust, improved patient satisfaction, and greater societal acceptance of AI-driven healthcare

Notes: These recommendations are grounded in a review of the current literature and the authors’ professional experiences in healthcare, AI ethics, and policy. They serve as an initial framework for stakeholders to collaboratively refine and implement strategies that ensure AI in healthcare remains trustworthy, ethical, and sustainable over the long term.

**Table 5 jcm-14-01605-t005:** Quantifiable framework for Trustworthy AI in healthcare.

Dimension	Key Metrics	Examples/Implications	Reference(s)
Transparency	Explainability Score, % of AI decisions with interpretable outputs	Feature attribution techniques such as SHAP or LIME are used to justify model predictions in clinical decision-making	Vimbi et al., 2024 [119]
Fairness	Bias reduction rate, model performance disparity across demographics	AI models benchmarked across diverse skin tones to ensure equitable diagnosis accuracy	Daneshjou et al., 2022 [120]
Accountability	Compliance rate with AI ethics guidelines, human-in-the-loop ratio	AI-assisted radiology systems requiring clinician oversight before issuing diagnostic reports	Adler-Milstein et al., 2022 [121]
Safety	Error rate reduction, model robustness under adversarial testing	Stress-testing AI-driven clinical decision tools to minimize false positives and negatives	Finlayson et al., 2019 [45]
Sustainability	Carbon footprint of AI training, green AI adoption rate	Evaluating computational costs and implementing energy-efficient AI models in hospitals	Dewasiri et al., 2025 [122]; Richie, 2022 [123]

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
