# Peer review of "Shaping the Future of Healthcare: Ethical Clinical Challenges and Pathways to Trustworthy AI"

_jcm, 2025, doi:10.3390/jcm14051605_

Round 1

Reviewer 1 Report

Comments and Suggestions for Authors

Review Report

1.      Summary

The manuscript titled "Shaping the Future of Healthcare: Ethical Clinical Challenges and Pathways to Trustworthy AI" offers a forward-looking perspective on the integration of AI into healthcare. It proposes novel concepts, including the "regulatory genome," and emphasizes transparency, accountability, sustainability, and global collaboration. The article identifies gaps in current ethical and governance frameworks and suggests pathways for addressing these challenges.

2.      General Concept Comments

In terms of strengths, the manuscript provides a timely and forward-looking perspective on the integration of AI into healthcare, addressing key ethical and governance challenges. The "regulatory genome" concept is innovative and has the potential to offer a structured approach to trustworthy AI development. The emphasis on sustainability and cross-disciplinary collaboration reflects a nuanced understanding of the broader implications of AI in healthcare. The manuscript is clear, well-structured, and relevant to the journal’s audience. It effectively highlights the urgency of addressing ethical challenges in healthcare AI.

As for weakness and suggestions:

·       The manuscript references relevant recent literature but lacks sufficient engagement with other frameworks addressing similar topics. For example, the discussion on bias and fairness (Lines 200–218) would benefit from citing studies demonstrating practical methods for reducing algorithmic bias in healthcare AI systems. Add a comparative discussion of existing frameworks or reviews to better situate the manuscript's contributions in the current research landscape.

·       While the manuscript is a perspective article, it does not clearly outline how the references were selected or synthesized. This reduces transparency and may limit the perceived rigor of the arguments. The authors are suggested including a brief description of the criteria used to identify and select relevant studies or examples to enhance methodological transparency.

·       The "regulatory genome" is introduced as a novel idea, but it remains largely theoretical. There are no examples or scenarios illustrating how it could be implemented in healthcare practice. Providing a case study or hypothetical example demonstrating the utility and feasibility of the "regulatory genome" in a clinical setting would add more

·       The figures and tables, particularly Table 1 and Figure 2, are useful but underexplained in the text. For example, Figure 2 lacks details about how the axes are defined and how stakeholders can apply the framework.

·       The manuscript's key arguments and frameworks are clear but could be enhanced with practical examples and deeper engagement with the figures and tables. For instance, Figure 2 would benefit from an expanded explanation in the text.

·       The conclusions align with the arguments presented but could be strengthened by summarizing actionable recommendations or proposing concrete next steps for research or policy.

·       The English is clear and academically appropriate. However, minor improvements (e.g., simplifying dense sentences in the abstract and introduction) could enhance overall readability.

Specific Comments

  • Lines 35–44: Add a concise statement of the research gap to contextualize the need for this article.
  • Line 49: The term "opaque 'black boxes'" could be clarified further by referencing specific examples of AI applications in healthcare where transparency has been a challenge.
  • Lines 216–218: Offer actionable recommendations for mitigating bias, such as partnerships with diverse communities or targeted data curation initiatives.
  • Transparency and Explainability (Line 330): Streamline repetitive discussions on the "black-box problem" to avoid redundancy with earlier sections.
  • Lines 346–350: Provide examples of "explainability scores" and how they can be integrated into clinical workflows.
  • Figure 2 (Line 320): Add a detailed description of how stakeholders can use the axes of the framework in practice.
  • Cross-Disciplinary Collaboration (Lines 464–469): Include specific examples of successful international collaborations (e.g., ISO or IEEE standards).
  • Line 488-489: Support the statement "international consortia can refine these into workable standards" with references or examples.
  • Lines 536–540: Expand on the concept of "green AI" with examples of tools or metrics to measure AI’s environmental impact.
  • Lines 567–580: Strengthen the conclusion by including a call for empirical research to validate the proposed frameworks.

Author Response

Response to Reviewer-1 Comments:

Thank you for your insightful review of our manuscript. We greatly appreciate your time and constructive feedback aimed at improving the clarity and quality of our work.

In response to the insights provided, we have undertaken thorough revisions to the manuscript, highlighting the modified sections in red color for your convenience.

Our responses to your comments are detailed below in a point-by-point manner:

Comment

Response

1.     Summary

The manuscript titled "Shaping the Future of Healthcare: Ethical Clinical Challenges and Pathways to Trustworthy AI" offers a forward-looking perspective on the integration of AI into healthcare. It proposes novel concepts, including the "regulatory genome," and emphasizes transparency, accountability, sustainability, and global collaboration. The article identifies gaps in current ethical and governance frameworks and suggests pathways for addressing these challenges.

We sincerely appreciate the reviewer’s thoughtful summary and recognition of our manuscript’s contributions. Our goal was to provide a forward-looking perspective on AI in healthcare while introducing novel concepts such as the “regulatory genome” and emphasizing the importance of transparency, accountability, sustainability, and global collaboration. We are pleased that the reviewer acknowledges our efforts in identifying gaps in existing ethical and governance frameworks and proposing pathways to address them.

Thank you for your valuable feedback.

2.     General Concept Comments

In terms of strengths, the manuscript provides a timely and forward-looking perspective on the integration of AI into healthcare, addressing key ethical and governance challenges. The "regulatory genome" concept is innovative and has the potential to offer a structured approach to trustworthy AI development. The emphasis on sustainability and cross disciplinary collaboration reflects a nuanced understanding of the broader implications of AI in healthcare. The manuscript is clear, well-structured, and relevant to the journal’s audience. It effectively highlights the urgency of addressing ethical challenges in healthcare AI.

We are grateful for your positive feedback and thoughtful evaluation of our manuscript. We appreciate your recognition of the timely and forward-looking perspective we aim to offer regarding AI integration into healthcare. The endorsement of our “regulatory genome” concept as an innovative approach to structuring trustworthy AI development is particularly encouraging.

Your acknowledgment of our emphasis on sustainability and cross-disciplinary collaboration as essential elements highlights our commitment to addressing the broader implications of AI in healthcare. We are pleased that our manuscript’s clarity, structure, and relevance to the journal’s audience resonate with your evaluation, particularly the urgency of tackling ethical challenges in healthcare AI.

Thank you once again for your insightful comments.

As for weakness and suggestions:

The manuscript references relevant recent literature but lacks sufficient engagement with other frameworks addressing similar topics. For example, the discussion on bias and fairness (Lines 200–218) would benefit from citing studies demonstrating practical methods for reducing algorithmic bias in healthcare AI systems. Add a comparative discussion of existing frameworks or reviews to better situate the manuscript's contributions in the current research landscape

We appreciate the reviewer’s feedback on the need for practical measures to address data silos and biases in AI healthcare applications. In response, we have expanded our discussion on bias mitigation strategies in Section 2.2.2 Mitigating Bias through Diverse Datasets and Algorithmic Audits. This section now provides concrete methods to reduce bias, including:

  • Collaborative data collection strategies with patient advocacy groups and community health organizations.
  • Standardized data curation protocols to ensure balanced representation across demographic groups.
  • Algorithmic fairness techniques such as adversarial debiasing and differential model tuning.
  • Regular fairness audits and real-world performance evaluations to detect and correct bias.
  • Independent validations by external researchers or institutions to ensure model fairness and reliability.
  • Policy-driven accountability mechanisms, including mandated transparency reporting and third-party audits.

These additions ensure that our discussion is grounded in actionable solutions rather than solely identifying the problem. Thank you for this valuable suggestion, and we hope these changes address your concern.

While the manuscript is a perspective article, it does not clearly outline how the references were selected or synthesized. This reduces transparency and may limit the perceived rigor of the arguments. The authors are suggested to include a brief description of the criteria used to identify and select relevant studies or examples to enhance methodological transparency.

We sincerely appreciate the reviewer’s valuable feedback on the need for greater transparency in our selection and synthesis of references. In response, we have revised the manuscript to explicitly outline the methodology used to identify, evaluate, and integrate relevant literature. Specifically, we have:

  • Clarified our reference selection criteria in the Abstract, Introduction, Section 3.1, and Conclusion, emphasizing the inclusion of peer-reviewed journal articles in PubMed, Scopus and Web of Science, published within the last decade.
  • Detailed our approach to literature synthesis, ensuring that selected sources provide a balanced perspective on AI ethics, regulatory adaptation, bias mitigation, and sustainability in healthcare.
  • Enhanced methodological transparency by integrating explicit justifications for reference selection within key sections discussing AI governance, explainability techniques, and interdisciplinary collaboration.

We appreciate the reviewer’s thoughtful suggestion and believe these changes strengthen the overall contribution of our work.

Thank you for your insightful feedback.

The "regulatory genome" is introduced as a novel idea, but it remains largely theoretical. There are no examples or scenarios illustrating how it could be implemented in healthcare practice. Providing a case study or hypothetical example demonstrating the utility and feasibility of the "regulatory genome" in a clinical setting would add more

We appreciate the reviewer’s insightful comment on the need for a more concrete illustration of the “regulatory genome” concept. To address this, we have expanded our discussion by incorporating a hypothetical case study that demonstrates how the regulatory genome can be implemented in healthcare AI governance. Specifically, we present a scenario regarding AI-Driven Early Detection of Diabetic Retinopathy in Low-Resource Settings.

These additions provide tangible insights into how the regulatory genome could function in real-world applications, addressing the theoretical nature of the concept and reinforcing its practical feasibility in AI governance. We have highlighted these updates in Section 1.5. Towards a Regulatory Genome of the revised manuscript.

The figures and tables, particularly Table 1 and Figure 2, are useful but underexplained in the text. For example, Figure 2 lacks details about how the axes are defined and how stakeholders can apply the framework.

We appreciate the reviewer’s feedback on the need for further explanation of Table 1 and Figure 2 in the text. In response, we have made the following revisions to enhance clarity and usability:

  1. Table 1 – We have expanded the discussion in Section 2.1 to explicitly explain the key dimensions outlined in the table. Each aspect, such as sensitive data handling, regulatory gaps, and technological risks, is now better integrated into the main text with additional context and real-world implications.
  2. Figure 2 – We have revised Section 3 to provide a more detailed explanation of the figure, including:
    • Definition of the axes – The horizontal axis (Ease of Integration) differentiates helpful internal attributes that enhance AI adoption from harmful external factors that hinder implementation. The vertical axis (Impact on Outcomes) distinguishes between internal organizational efforts and external regulatory, ethical, and societal constraints.
    • Stakeholder application – We now clarify how different AI models or initiatives can be positioned within the framework and how stakeholders (e.g., AI developers, healthcare providers, regulators) can use it to assess and refine their AI systems.

We appreciate the reviewer’s suggestion, which has strengthened the manuscript’s clarity.

Thank you for your insightful feedback.

The manuscript's key arguments and frameworks are clear but could be enhanced with practical examples and deeper engagement with the figures and tables. For instance, Figure 2 would benefit from an expanded explanation in the text.

Thank you for your valuable feedback. We have expanded the discussion of Figure 2 by adding examples (in Section 3) to illustrate AI model placement within the framework. Specifically, we now:

  1. Clarify how rule-based AI, deep learning, and hybrid models align with trust, accountability, and explainability.
  2. Provide real-world examples, such as deep learning for tumor detection, rule-based systems for clinical decision support, and hybrid AI in diabetic retinopathy.
  3. Highlight how organizations can assess AI maturity and align deployments with privacy, compliance, and sustainability goals.

The conclusions align with the arguments presented but could be strengthened by summarizing actionable recommendations or proposing concrete next steps for research or policy

We appreciate the reviewer’s valuable suggestion to strengthen the conclusion by summarizing actionable recommendations and concrete next steps for research and policy. In response, we have revised the conclusion to include:

  1. Actionable Recommendations: We now explicitly outline key areas for improvement, including regulatory adaptation, interdisciplinary cooperation, global policy alignment, and priority research areas. These provide clear directions for stakeholders involved in AI governance and implementation.
  2. Concrete Next Steps for Research & Policy: We highlight the need for flexible regulatory frameworks, empirical evaluations of AI fairness, real-world applications of AI transparency, and sustainability metrics. These recommendations aim to guide future work in both academic and policy domains.

The conclusion not only aligns with the arguments presented in the manuscript but also offers a forward-looking perspective that can inform practical advancements in AI ethics and governance.

Thank you for your insightful feedback.

The English is clear and academically appropriate. However, minor improvements (e.g., simplifying dense sentences in the abstract and introduction) could enhance overall readability

We appreciate the reviewer’s feedback on improving readability and simplifying dense sentences in the abstract and introduction. In response, we have revised these sections to enhance clarity and conciseness while maintaining academic rigor. Specifically, we have:

  1. Simplified sentence structures to improve readability without compromising the depth of analysis.
  2. Clarified key arguments related to AI’s ethical, regulatory, and societal challenges, ensuring a smoother flow between concepts.
  3. Refined complex sentences, such as restructuring discussions on AI’s role in addressing health disparities, making them more direct and impactful.

Thank you for your valuable feedback.

Lines 35–44: Add a concise statement of the research gap to contextualize the need for this article.

We appreciate the reviewer’s suggestion to clarify the research gap and contextualize the need for this article. In response, we have revised Lines 35–44 to explicitly state the existing gap in AI ethics, governance, and implementation research. The updated text now highlights how previous studies often examine AI’s technological advancements in isolation while overlooking the intersection of ethical risks, regulatory gaps, and real-world deployment challenges.

To address this gap, the article presents a multidisciplinary framework integrating perspectives from healthcare, ethics, policy, and AI governance, offering concrete strategies for trustworthy AI implementation.

We appreciate the reviewer’s insightful feedback, which has strengthened the clarity and impact of the manuscript.

Thank you for your valuable suggestions.

Line 49: The term "opaque 'black boxes'" could be clarified further by referencing specific examples of AI applications in healthcare where transparency has been a challenge.

Thank you for your insightful comment. We appreciate the opportunity to enhance the clarity of our discussion on the challenges associated with opaque AI models in healthcare. In response, we have revised the text to provide concrete examples of AI applications where transparency has been a significant issue.

Specifically, we have included references in Section 1.1. The Promise and Risks of AI in Healthcare.

Thank you for your valuable feedback.

Lines 216–218: Offer actionable recommendations for mitigating bias, such as partnerships with diverse communities or targeted data curation initiatives.

Thank you for your valuable suggestion. We have revised Section 2.2.2 to include specific, actionable strategies for mitigating bias in AI-driven healthcare. These include:

  • Partnerships with diverse communities to ensure inclusive data collection and validation.
  • Targeted data curation initiatives to improve dataset diversity.
  • Bias-aware model development, integrating fairness techniques like adversarial debiasing and reweighting.
  • Continuous auditing and performance monitoring across demographic groups.
  • Regulatory incentives requiring transparency, fairness audits, and demographic-specific performance reporting.

These additions enhance practical applicability and ensure a proactive approach to bias mitigation. We appreciate your insightful feedback.

Transparency and Explainability (Line 330): Streamline repetitive discussions on the "black-box problem" to avoid redundancy with earlier sections.

Thank you for your feedback. In response, we have streamlined the discussion on the "black-box problem" to avoid redundancy with earlier sections. The revised text now focuses on practical solutions, emphasizing interpretable model architectures, explainability techniques, and user-friendly decision-support tools. Additionally, we have maintained the relevance of transparency in AI adoption while ensuring a more concise and structured discussion.

We appreciate your insightful suggestion, which has helped improve the clarity and coherence of this section.

Lines 346–350: Provide examples of "explainability scores" and how they can be integrated into clinical workflows.

Thank you for your insightful comment. We have now expanded the section on Proposing Measurable Metrics for Transparency in AI Systems to include specific examples of explainability scores and their practical integration into clinical workflows. The revised section highlights several widely used techniques for quantifying AI explainability, including SHapley Additive exPlanations (SHAP), Local Interpretable Model-agnostic Explanations (LIME), and attention mechanisms in neural networks.

We appreciate your valuable feedback, which has helped us improve the practical relevance of this discussion.

Figure 2 (Line 320): Add a detailed description of how stakeholders can use the axes of the framework in practice.

We appreciate the reviewer’s feedback on the need for further explanation of Figure 2 in the text.

We now clarify how different AI models or initiatives can be positioned within the framework and how stakeholders (e.g., AI developers, healthcare providers, regulators) can use it to assess and refine their AI systems.

We have revised Section 3 to provide a more detailed explanation of the figure.

Thank you for your insightful feedback

Cross-Disciplinary Collaboration (Lines 464–469): Include specific examples of successful international collaborations (e.g., ISO or IEEE standards).

Thank you for your insightful suggestion. In response, we have incorporated specific examples of successful international collaborations, such as ISO/IEC 42001:2023, which provides a structured AI management framework, and IEEE 2801-2022, which ensures quality management for AI-driven medical datasets. These examples highlight the role of global standardization efforts in harmonizing AI governance across jurisdictions.

Additionally, we have clarified the role of industry-government partnerships, emphasizing how international consortia, regulatory bodies, and professional organizations refine baseline policies into practical governance models aligned with global standards and the UN Sustainable Development Goals (SDGs).

We appreciate your valuable feedback, which has helped improve the clarity and depth of this section.

Line 488-489: Support the statement "international consortia can refine these into workable standards" with references or examples.

We have provided concrete examples of international consortia actively contributing to the refinement of AI standards into practical, implementable frameworks. Specifically, we have incorporated references to:

  1. SPIRIT-AI and CONSORT-AI Guidelines – These provide structured international standards for the design, reporting, and evaluation of clinical trials involving AI interventions. By standardizing clinical trial protocols, these guidelines help translate broad regulatory principles into actionable methodologies for AI in healthcare.
  2. FUTURE-AI Consortium – This initiative has established a globally recognized framework for trustworthy and deployable AI in healthcare, ensuring compliance with both technical and ethical considerations.

These examples demonstrate how international collaborations not only set overarching guidelines but also refine them into domain-specific, operational standards.

We appreciate the reviewer’s suggestion, which has strengthened the clarity and evidential support for this section.

Lines 536–540: Expand on the concept of "green AI" with examples of tools or metrics to measure AI’s environmental impact.

Thank you for your valuable suggestion. In response, we have expanded the discussion on “green AI” by incorporating specific tools and metrics used to assess AI’s environmental impact. We have added references to Carbontracker (Anthony et al., 2020), which measures the carbon footprint of AI training, and the Energy Usage Reports framework (Lottick et al., 2019), which advocates for energy transparency in machine learning models. These additions provide concrete examples of how AI’s energy consumption can be quantified and mitigated.

This revision enhances the practical applicability of the discussion, ensuring that stakeholders have actionable tools to implement sustainability-focused AI development. We appreciate your insightful feedback, which has strengthened the section’s clarity and impact.

Lines 567–580: Strengthen the conclusion by including a call for empirical research to validate the proposed frameworks.

We appreciate the reviewer’s valuable suggestion to strengthen the conclusion by summarizing actionable recommendations and concrete next steps. In response, we have revised the conclusion to include:

  1. Actionable Recommendations: We now explicitly outline key areas for improvement, including regulatory adaptation, interdisciplinary cooperation, global policy alignment, and priority research areas. These provide clear directions for stakeholders involved in AI governance and implementation.
  2. Concrete Next Steps: We highlight the need for flexible regulatory frameworks, empirical evaluations of AI fairness, real-world applications of AI transparency, and sustainability metrics. These recommendations aim to guide future work in both academic and policy domains.

The conclusion not only aligns with the arguments presented in the manuscript but also offers a forward-looking perspective that can inform practical advancements in AI ethics and governance.

Thank you for your insightful feedback.

Reviewer 2 Report

Comments and Suggestions for Authors

The subject is significant. Nevertheless, I possess many apprehensions:

1.      The introduction highlights AI's ability to resolve enduring difficulties. Could the authors provide explicit instances of these issues and delineate how AI effectively mitigates them?

2.      The paper addresses the opaque characteristics of artificial intelligence. Although this is a prevalent issue, are specific categories of AI used in healthcare intrinsically superior to others? This needs further nuance.

3.      The discourse on data privacy and cyberattacks might be enhanced by referencing instances of breaches or weaknesses inside healthcare AI systems. The authors have to provide citations and elaborate on their results.

4.      The text refers to regulatory uncertainty. Could the authors provide explicit instances of this uncertainty, comparing regional methodologies or emphasizing sectors where regulation is absent?

5.      The introduction addresses ethical, regulatory, and social issues. Although moral and regulatory concerns are thoroughly examined, social considerations are not discussed in length. Could the authors elaborate on these social implications?

6.      Trustworthy AI encompasses openness, accountability, justice, and patient autonomy. In addition to these, might other pertinent concepts, such as safety, security, or robustness, be considered? Thus, the authors might consider these issues too in their discussion.

7.      The essay highlights the need for representative and varied training data to ensure fairness. What practical measures may be implemented in healthcare to address the challenges posed by data silos and biases in current datasets?

8.      The essay asserts that attaining reliable AI is a multidisciplinary task. How does the study explicitly tackle the interdisciplinary dimension within its recommended framework or recommendations?

9.      The notion of AI as an ecosystem is presented. In what manner does this ecosystem viewpoint directly influence the subsequent arguments or suggestions presented in the paper? Is this just a framing mechanism, or does it provide substantive insights?

10.  The paper references the EU's AIA. How does this legislation contrast with other global regulatory approaches? A concise comparison study may be advantageous.

11.  The notion of a "regulatory genome" is presented. This idea is intriguing, but how is it implemented? What are the tangible ramifications of this concept? In what ways does it diverge from current regulatory methodologies?

12.  The paper refers to aligning AI development with the United Nations Sustainable Development Goals (SDGs). Although this objective is commendable, what methods may be used to assess or implement it effectively?

13.  The language references adversarial assaults on AI models. Could the authors provide explicit instances of such assaults inside the healthcare sector and examine their possible ramifications?

14.  The paper addresses the improper use of generative AI techniques. Could authors provide explicit instances of abuse within the healthcare domain and examine their possible repercussions?

15.  The paper asserts that it fills a vacuum in the literature by offering a cohesive methodology. In what ways does this approach diverge from current frameworks or guidelines? What is the distinctive contribution of this paper?

16.  The document offers "quantifiable frameworks and policy suggestions." These are not immediately evident in the supplied snippets. I would appreciate a distinct section with appropriate examples. The paper's contribution may be improved.

17.  The conclusion addresses the integration of theoretical ideas with actual applications. The authors must provide explicit examples in the manuscript of how they address this gap.

18.  The paper highlights "emerging challenges including adversarial resilience and sustainable AI development." These subjects are significant. The authors should explain their development process using a pertinent example.

Comments on the Quality of English Language

A minor modification in English enhances the paper's readability.

Author Response

Response to Reviewer-2 Comments:

Thank you for your thoughtful and detailed review of our manuscript. We sincerely appreciate the time and effort you invested in providing valuable feedback to enhance the clarity and overall quality of our work.

In light of your comments, we have carefully revised the manuscript, with all modifications clearly highlighted for ease of reference.

Below, we provide a point-by-point response to each of your comments:

Comment

Response

The subject is significant. Nevertheless, I possess many apprehensions:

 1. The introduction highlights AI's ability to resolve enduring difficulties. Could the authors provide explicit instances of these issues and delineate how AI effectively mitigates them?

We appreciate the reviewer’s feedback on the need for practical measures to address data silos and biases in AI healthcare applications. In response, we have expanded our discussion on bias mitigation strategies in Section 2.2.2 Mitigating Bias through Diverse Datasets and Algorithmic Audits. This section now provides concrete methods to reduce bias, including:

  • Collaborative data collection strategies with patient advocacy groups and community health organizations.
  • Standardized data curation protocols to ensure balanced representation across demographic groups.
  • Algorithmic fairness techniques such as adversarial debiasing and differential model tuning.
  • Regular fairness audits and real-world performance evaluations to detect and correct bias.
  • Independent validations by external researchers or institutions to ensure model fairness and reliability. 
  • Policy-driven accountability mechanisms, including mandated transparency reporting and third-party audits.

We have also expanded the discussion of Figure 2 by adding examples (in Section 3) to illustrate AI model placement within the framework. Specifically, we now:

  1. Clarify how rule-based AI, deep learning, and hybrid models align with trust, accountability, and explainability.
  2. Provide real-world examples, such as deep learning for tumor detection, rule-based systems for clinical decision support, and hybrid AI in diabetic retinopathy.

These additions ensure that our discussion is grounded in actionable solutions rather than solely identifying the problem. Thank you for this valuable suggestion, and we hope these changes address your concern.

2. The paper addresses the opaque characteristics of artificial intelligence. Although this is a prevalent issue, are specific categories of AI used in healthcare intrinsically superior to others? This needs further nuance.

We appreciate the reviewer’s insightful comment on the need for further nuance in discussing different AI categories in healthcare. In response, we have expanded the Introduction & Section 3 in the manuscript to highlight the comparative strengths and limitations of various AI models. Specifically, we have clarified that AI systems differ in their suitability for healthcare applications based on their interpretability, complexity, and predictive performance.

To address this, we now emphasize that while symbolic AI and rule-based expert systems offer greater explainability, they struggle with handling complex, large-scale medical data. In contrast, deep learning models provide higher accuracy for image analysis and pattern recognition but pose interpretability challenges. Furthermore, hybrid AI models, which integrate symbolic reasoning with machine learning, have emerged as promising solutions for balancing transparency and performance in clinical decision-making.

Additionally, we have refined the explanation of how AI selection should align with specific healthcare applications. For example, deep learning models may be more suitable for complex imaging tasks such as tumor detection, while rule-based systems are preferable for clinical decision support where interpretability and transparency are paramount. We also emphasize the importance of explainable AI (XAI) techniques as a bridge between accuracy and interpretability.

We hope these additions adequately address the reviewer’s concerns, and we welcome any further suggestions.

3. The discourse on data privacy and cyberattacks might be enhanced by referencing instances of breaches or weaknesses inside healthcare AI systems. The authors have to provide citations and elaborate on their results.

Thank you for your valuable feedback. We appreciate your suggestion to enhance the discussion on data privacy and cybersecurity risks by referencing real-world instances of breaches or vulnerabilities within healthcare AI systems. In response, we have incorporated examples of significant cybersecurity incidents in the healthcare sector, including the 2015 Anthem data breach, which compromised the personal data of 78.8 million individuals, and the 2021 ransomware attack on Ireland’s Health Service Executive (HSE), which disrupted nationwide healthcare services. Additionally, we have referenced research highlighting the risks posed by cyber threats to electronic health records (EHRs) and cloud-based healthcare AI infrastructure. These additions strengthen our argument regarding the critical need for robust cybersecurity measures, encryption protocols, and real-time threat detection strategies in AI-driven healthcare.

Thank you again for your thoughtful review and for helping us strengthen our discussion.

4. The text refers to regulatory uncertainty. Could the authors provide explicit instances of this uncertainty, comparing regional methodologies or emphasizing sectors where regulation is absent?

Thank you for your valuable comment. We have revised the manuscript to provide explicit examples of regulatory uncertainty, comparing regional methodologies and highlighting sectors where regulation remains absent. Specifically, we have expanded our discussion in Section 2.3.1 Limitations of Current Regulatory Frameworks to illustrate how discrepancies between regulatory frameworks create uncertainty for healthcare AI implementation.

5. The introduction addresses ethical, regulatory, and social issues. Although moral and regulatory concerns are thoroughly examined, social considerations are not discussed in length. Could the authors elaborate on these social implications?

We appreciate the reviewer’s valuable feedback regarding the need for a more comprehensive discussion of the social implications of AI in healthcare. To address this concern, we have expanded the Introduction section to explicitly discuss the social impact of AI integration in medical practice. Specifically, we have added new content discussing the digital divide, workforce adaptation, patient autonomy, and shifts in physician-patient interactions due to AI-assisted clinical decision-making.

Thank you for the insightful suggestion.

6. Trustworthy AI encompasses openness, accountability, justice, and patient autonomy. In addition to these, might other pertinent concepts, such as safety, security, or robustness, be considered? Thus, the authors might consider these issues too in their discussion.

We appreciate the reviewer’s insightful suggestion regarding additional dimensions of trustworthy AI. In response, we have expanded our discussion to explicitly include safety, security, and robustness as essential pillars of trustworthy AI in healthcare.

  • Safety is crucial to ensuring that AI-driven systems prioritize patient well-being and do not introduce unintended risks in clinical settings. We have now incorporated relevant references highlighting patient safety concerns related to AI-powered decision support systems and the need for rigorous validation before deployment (Bates et al., 2021).
  • Security plays a vital role in safeguarding sensitive health data against cyber threats and unauthorized access. Given the increasing frequency of healthcare-related cyberattacks, AI systems must integrate strong encryption, access controls, and real-time anomaly detection mechanisms (Layode et al., 2024).
  • Robustness ensures that AI systems function reliably across diverse patient populations and clinical environments. This includes resilience against adversarial attacks, handling of unforeseen variations in real-world data, and ensuring that AI performance remains consistent across demographic groups (Mittelstadt, 2019).

We have revised the manuscript accordingly (see Section 1.2: Defining Trustworthy AI).

We appreciate the reviewer’s valuable feedback, which has helped us refine our argument and provide a more holistic framework for trustworthy AI in healthcare.

7. The essay highlights the need for representative and varied training data to ensure fairness. What practical measures may be implemented in healthcare to address the challenges posed by data silos and biases in current datasets?

We appreciate the reviewer's insightful comment regarding the need for practical solutions to mitigate data silos and biases in AI-driven healthcare applications. In response, we have added Section 4.4 (Establishing Quantifiable AI Governance Metrics) under the Fairness dimension to include specific strategies addressing these challenges.

8. The essay asserts that attaining reliable AI is a multidisciplinary task. How does the study explicitly tackle the interdisciplinary dimension within its recommended framework or recommendations?

Thank you for this insightful comment. We agree that achieving reliable AI in healthcare necessitates an interdisciplinary approach, and we appreciate the opportunity to clarify how this study explicitly integrates multiple disciplines within its framework and recommendations.

In our revised manuscript, we have strengthened the discussion on interdisciplinary collaboration by explicitly outlining how contributions from healthcare professionals, AI engineers, ethicists, legal experts, and policymakers collectively shape AI governance. Specifically, we have:

  1. Expanded the Introduction to emphasize the necessity of interdisciplinary perspectives in addressing AI's ethical, regulatory, and implementation challenges. We now explicitly state how legal, sociological, technological, and public health viewpoints inform our synthesis.
  2. Revised the Trustworthy AI Section (1.3) to illustrate how different disciplinary expertise contributes to key AI attributes, such as explainability (computer science and healthcare), regulatory compliance (law and public policy), and ethical considerations (bioethics and philosophy).
  3. Enhanced the Governance Framework Section (4.1) to highlight how our recommendations are built on cross-disciplinary collaborations. We now provide specific examples of how AI regulatory adaptation strategies benefit from inputs across healthcare, law, and ethics, ensuring practical feasibility.

We appreciate the reviewer’s valuable suggestion, as it has helped refine our argument and strengthen the manuscript’s clarity.

9. The notion of AI as an ecosystem is presented. In what manner does this ecosystem viewpoint directly influence the subsequent arguments or suggestions presented in the paper? Is this just a framing mechanism, or does it provide substantive insights?

We appreciate the reviewer’s insightful question regarding how the AI as an ecosystem perspective influences our subsequent arguments and suggestions. This concept is not merely a framing mechanism but a foundational lens that shapes our approach to AI governance, ethical considerations, and regulatory recommendations.

To clarify this connection, we emphasize that viewing AI as an ecosystem emphasizes the interdependence of AI technologies, regulatory structures, clinical workflows, and societal factors (Section 1.4, 1.5 & 1.6)—a theme that is carried throughout the paper. Specifically:

  • Regulatory Genome: The ecosystem perspective directly informs our proposal for a regulatory genome, highlighting that AI governance must be dynamic, adaptive, and inclusive of diverse stakeholders to function effectively within evolving healthcare environments.
  • SDG Alignment & AI Governance Audits: The emphasis on AI’s systemic impact translates into concrete policy recommendations such as impact assessment metrics and AI governance audits. These measures ensure AI solutions are aligned with global sustainability goals and effectively integrated into healthcare infrastructures.
  • Bias Mitigation and Data Governance: Recognizing AI as an interconnected system reinforces our argument for comprehensive data governance strategies. Addressing bias is not an isolated technical challenge but requires system-wide interventions, including standardized data protocols, interdisciplinary collaboration, and fairness-driven regulatory policies.

By integrating AI as an ecosystem into our analytical framework, our recommendations go beyond abstract principles and offer actionable strategies for interoperability, equity, sustainability, and regulatory alignment. To further strengthen this connection, we have clarified these links within the manuscript to ensure that the ecosystem perspective remains a guiding principle rather than a standalone concept.

10. The paper references the EU's AIA. How does this legislation contrast with other global regulatory approaches? A concise comparison study may be advantageous.

Thank you for your insightful comment regarding the comparison of the EU’s Artificial Intelligence Act (AIA) with other global regulatory frameworks. In response, we have expanded the discussion to provide a comparative analysis of AI regulations in different jurisdictions. Specifically, we now highlight how the AIA emphasizes transparency, risk assessment, and documentation but lacks clear post-market adaptive oversight mechanisms. This contrasts with the UK’s NHS AI regulations, which offer a more flexible, real-world evidence-based approach, albeit without legally binding mandates. Additionally, we discuss the regulatory uncertainties in the U.S., where the FDA has yet to establish a standardized framework for continuously learning AI systems. This comparison emphasizes the challenges AI developers face when navigating diverse regulatory landscapes, ultimately impacting global AI adoption in healthcare.

We believe this addition strengthens the manuscript by contextualizing the AIA within a broader regulatory framework and demonstrating how differing approaches impact AI deployment. We appreciate the reviewer’s suggestion, which has helped refine our discussion.

11. The notion of a "regulatory genome" is presented. This idea is intriguing, but how is it implemented? What are the tangible ramifications of this concept? In what ways does it diverge from current regulatory methodologies?

We appreciate the reviewer’s insightful comment on the need for a more concrete illustration of the “regulatory genome” concept. To address this, we have expanded our discussion by incorporating a hypothetical case study that demonstrates how the regulatory genome can be implemented in healthcare AI governance. Specifically, we present a scenario regarding AI-Driven Early Detection of Diabetic Retinopathy in Low-Resource Settings.

These additions provide tangible insights into how the regulatory genome could function in real-world applications, addressing the theoretical nature of the concept and reinforcing its practical feasibility in AI governance. We have highlighted these updates in Section 1.5. Towards a Regulatory Genome of the revised manuscript.

12. The paper refers to aligning AI development with the United Nations Sustainable Development Goals (SDGs). Although this objective is commendable, what methods may be used to assess or implement it effectively?

We appreciate the reviewer's valuable insight regarding the need for concrete methods to assess and implement AI alignment with the SDGs. To address this, we have incorporated a structured framework that outlines key methods for evaluating AI’s contribution to sustainable development. These methods include:

  1. Impact Assessment Metrics – We propose developing standardized AI impact indicators aligned with specific SDGs, such as healthcare accessibility (SDG 3), reduced inequalities (SDG 10), and responsible innovation (SDG 9). These indicators allow for the quantification of AI-driven progress toward sustainable healthcare.
  2. AI Governance Audits – We suggest implementing periodic audits to evaluate AI systems in healthcare, ensuring they adhere to sustainability principles, ethical compliance, and fairness. This aligns with emerging AI auditing frameworks that assess adherence to global sustainability goals.
  3. Regulatory Incentives – We highlight the importance of encouraging AI developers and healthcare institutions to adopt SDG-aligned AI practices through regulatory incentives, including funding opportunities and compliance certifications that promote responsible AI innovation.
  4. Stakeholder-driven Evaluation – To ensure holistic assessment, we advocate for engaging multidisciplinary experts—including healthcare professionals, ethicists, and policymakers—in qualitative and quantitative reviews of AI’s contributions to sustainable healthcare ecosystems.

These additions have been integrated into the revised manuscript to strengthen the discussion on the practical implementation of SDG alignment in AI-driven healthcare. We appreciate the reviewer’s suggestion, which has significantly enhanced the clarity and applicability of our argument.

13. The language references adversarial assaults on AI models. Could the authors provide explicit instances of such assaults inside the healthcare sector and examine their possible ramifications?

Thank you for your valuable comment regarding adversarial assaults on AI models. In response, we have expanded the manuscript by incorporating additional details in the sections,

  • Section 1.3. Defining Trustworthy AI,
  • Section 1.5. Towards a Regulatory Genome,
  • Section 2.2.2 Mitigating Bias through Diverse Datasets and Algorithmic Audits,

We appreciate the reviewer’s suggestion, which has helped us refine our discussion on AI security in healthcare.

14. The paper addresses the improper use of generative AI techniques. Could authors provide explicit instances of abuse within the healthcare domain and examine their possible repercussions?

We appreciate the reviewer’s insightful suggestion to include explicit instances of generative AI misuse in healthcare. In response, we have revised the manuscript to incorporate concrete examples supported by peer-reviewed literature. Specifically, we have detailed the following key areas of concern:

  1. Fabrication of Medical Records
  2. Misinformation in Medical Advice
  3. Algorithmic Bias
  4. Deepfake Medical Content

We have also expanded our discussion on the consequences of these abuses, emphasizing their impact on patient trust, clinical decision-making, and legal accountability. Additionally, we highlight the need for robust governance mechanisms, AI auditing, and interdisciplinary oversight to mitigate these risks effectively.

Thank you for your valuable feedback.

15. The paper asserts that it fills a vacuum in the literature by offering a cohesive methodology. In what ways does this approach diverge from current frameworks or guidelines? What is the distinctive contribution of this paper?

We appreciate the reviewer’s insightful question regarding the distinctive contribution of our paper and how our proposed methodology diverges from existing frameworks. To clarify, our approach moves beyond prior work by integrating a structured, interdisciplinary methodology that connects AI ethics, governance, and real-world implementation in healthcare, bridging the gap between theoretical principles and practical AI deployment strategies.

Key Contributions and Distinctions:

  1. Interdisciplinary Integration Beyond Existing AI Governance Frameworks
    • While previous frameworks often focus on either technical AI governance (e.g., explainability techniques, bias audits) or ethical AI principles (e.g., fairness, transparency, accountability), our paper synthesizes both by incorporating perspectives from computer science, healthcare policy, regulatory law, and social science.
    • Existing regulatory models (e.g., EU AI Act, FDA SaMD, NHS AI Lab) emphasize high-level compliance but often lack operationalizable, quantifiable governance metrics. Our AI Trustworthiness Framework (Fig. 2) systematically aligns fairness, transparency, sustainability, and regulatory adaptation, offering measurable implementation metrics (Table 5) that facilitate real-world AI assessment.
  2. Regulatory Genome: A Novel Adaptive Compliance Framework
    • Unlike static governance models, we introduce the “Regulatory Genome”, a dynamic AI governance approach that enables continuous monitoring, adaptive oversight, and stakeholder-driven regulatory evolution.
  3. Bridging AI Implementation Gaps: From Ethics to Practice
    • Existing ethical AI frameworks primarily provide abstract guidelines (e.g., IEEE AI Ethics, AI4People) but often lack enforceable mechanisms.
    • Our paper provides policy-driven strategies (Section 4.5) that operationalize AI fairness, explainability, and accountability into real-world governance mechanisms—such as mandatory bias audits, transparency reporting, and sustainability incentives.
    • This policy-to-practice transition is a critical advancement, ensuring AI is not just compliant but also effective, equitable, and socially responsible.

We added Section 1.7. Key Contributions of This Study in the manuscript.

By presenting a structured, cross-disciplinary, and policy-integrated framework, this paper fills a critical gap in AI governance by bridging ethical principles with real-world deployment strategies.

Thank you for your valuable feedback.

16. The document offers "quantifiable frameworks and policy suggestions." These are not immediately evident in the supplied snippets. I would appreciate a distinct section with appropriate examples. The paper's contribution may be improved.

We appreciate the reviewer’s suggestion to enhance the visibility of our quantifiable frameworks and policy suggestions. To address this, we have introduced a dedicated section (Section 4.4 & 4.5: Quantifiable Frameworks and Policy Recommendations) that explicitly outlines measurable indicators, performance metrics, and concrete policy strategies for trustworthy AI in healthcare.

In this new sections, we provide a structured framework that incorporates key performance metrics for AI governance, covering dimensions such as transparency, fairness, accountability, safety, and sustainability. Each dimension is supported by specific, quantifiable indicators (e.g., explainability scores, bias reduction rates, compliance percentages, and AI-related energy consumption metrics), ensuring an objective evaluation of AI impact in clinical settings.

Furthermore, we introduce policy recommendations that include regulatory adaptation policies, compliance incentives for sustainable AI, and AI auditing mechanisms to bridge the gap between ethical AI principles and real-world implementation.

17. The conclusion addresses the integration of theoretical ideas with actual applications. The authors must provide explicit examples in the manuscript of how they address this gap.

We appreciate the reviewer’s comment regarding the need for explicit examples that bridge theoretical ideas with real-world applications. To address this, we have incorporated hypothetical scenarios throughout the manuscript to illustrate how key concepts—such as the regulatory genome, AI trustworthiness, and ethical AI governance—can be applied in healthcare settings.

For instance, in Section 1.5. Towards a Regulatory Genome, we provide a hypothetical case demonstrating how an AI-powered clinical decision support system could be monitored and refined using the regulatory genome framework. This scenario illustrates the practical feasibility of an adaptive AI governance model that integrates real-time compliance updates, bias audits, and stakeholder feedback loops.

Similarly, in Section 3.1. Transparency and Explainability, we introduce practical examples of explainability techniques, such as SHAP (Shapley Additive Explanations) and LIME (Local Interpretable Model-Agnostic Explanations), showing how these methods enhance trust and usability in AI-driven diagnostics. Additionally, in Section 4.3. Sustainable and Ethical AI Development, we discuss AI’s environmental impact and propose structured metrics, such as Carbontracker, to assess the sustainability of AI models in healthcare.

These additions ensure that our discussion is not limited to abstract principles but is complemented by concrete, illustrative examples that demonstrate the real-world applicability of our proposed frameworks. We have highlighted these updates in the revised manuscript.

18. The paper highlights "emerging challenges including adversarial resilience and sustainable AI development." These subjects are significant. The authors should explain their development process using a pertinent example.

We appreciate the reviewer’s insightful suggestion to elaborate on the development process of adversarial resilience and sustainable AI development with specific examples. To address this, we have revised the relevant section to incorporate concrete illustrations.

Specifically, we now discuss adversarial attacks in medical AI, highlighting how subtle manipulations in input data can lead to incorrect diagnoses or treatment recommendations. We reference prior work (Finlayson et al., 2019) that demonstrates how adversarial perturbations can deceive deep learning models in clinical applications. Additionally, we emphasize the importance of developing robust AI models with adversarial training techniques and continuous monitoring mechanisms to enhance resilience.

Furthermore, in the context of sustainable AI development, we discuss the environmental impact of AI-driven healthcare solutions, particularly regarding computational power and energy consumption. We reference recent discussions on sustainable AI practices (Vishwakarma et al., 2025) that advocate for energy-efficient model architectures and responsible AI deployment strategies.

We appreciate your constructive feedback.

Round 2

Reviewer 1 Report

Comments and Suggestions for Authors

The revision addressed my comments and questions.